# Over-elongation of centrioles in cancer promotes centriole amplification and chromosome missegregation

Gaëlle Marteil[1], Adan Guerrero[1,11], André F. Vieira[2,3], Bernardo P. de Almeida[4,5], Pedro Machado[1,12], Susana Mendonça[1,2,3], Marta Mesquita[6], Beth Villarreal[7], Irina Fonseca[1], Maria E. Francia[1,13], Katharina Dores[1], Nuno P. Martins[8], Swadhin C. Jana[1], Erin M. Tranfield[9], Nuno L. Barbosa-Morais[4], Joana Paredes[2,3], David Pellman[10], Susana A. Godinho[10,14] & Mónica Bettencourt-Dias[1]

Centrosomes are the major microtubule organising centres of animal cells. Deregulation in their number occurs in cancer and was shown to trigger tumorigenesis in mice. However, the incidence, consequence and origins of this abnormality are poorly understood. Here, we screened the NCI-60 panel of human cancer cell lines to systematically analyse centriole number and structure. Our screen shows that centriole amplification is widespread in cancer cell lines and highly prevalent in aggressive breast carcinomas. Moreover, we identify another recurrent feature of cancer cells: centriole size deregulation. Further experiments demonstrate that severe centriole over-elongation can promote amplification through both centriole fragmentation and ectopic procentriole formation. Furthermore, we show that overly long centrioles form over-active centrosomes that nucleate more microtubules, a known cause of invasiveness, and perturb chromosome segregation. Our screen establishes centriole amplification and size deregulation as recurrent features of cancer cells and identifies novel causes and consequences of those abnormalities.

[1] Instituto Gulbenkian de Ciência, Oeiras 2780-156, Portugal. [2] I3S - Instituto de Investigação e Inovação em Saúde, Universidade do Porto, Porto 4200-135, Portugal. [3] IPATIMUP - Instituto de Patologia e Imunologia Molecular, Universidade do Porto, Porto 4200-135, Portugal. [4] Instituto de Medicina Molecular, Faculdade de Medicina, Universidade de Lisboa, Lisbon 1649-028, Portugal. [5] Departamento de Ciências Biomédicas e Medicina, Universidade do Algarve, Faro 8005-139, Portugal. [6] Instituto Português de Oncologia de Lisboa, Lisbon 1099-023, Portugal. [7] Novartis Institutes for BioMedical Research, Boston, MA 02139, USA. [8] Advanced Imaging Facility, Instituto Gulbenkian de Ciência, Oeiras 2780-156, Portugal. [9] Electron Microscopy Facility, Instituto Gulbenkian de Ciência, Oeiras 2780-156, Portugal. [10] Dana-Faber Cancer Institute, Boston, MA 02215-5450, USA. [11] Present address: Laboratorio Nacional de Microscopía Avanzada, Instituto de Biotecnología, Universidad Nacional Autónoma de México (UNAM), Cuernavaca, Morelos 62210, Mexico. [12] Present address: European Molecular Biology Laboratory, Heidelberg 69117, Germany. [13] Present address: Institut Pasteur de Montevideo, Montevideo 11400, Uruguay. [14] Present address: Molecular Oncology, Barts Cancer Institute, Queen Mary University of London, London EC1M 6BQ, United Kingdom. Correspondence and requests for materials should be addressed to G.M. (email: gaelle.marteil@gmail.com) or to M.B.-D. (email: mdias@igc.gulbenkian.pt)

Centrosomes are the major microtubule organising centres (MTOCs) of animal cells participating in signalling, cell division, polarity and migration[1–3]. Each centrosome comprises two centrioles surrounded by a proteinaceous matrix, the pericentriolar material (PCM), which confers the microtubule (MT) nucleation capacity[4]. Centrioles are microtubule-based cylinders and their structure, length (≈450 nm) and number (4 in mitosis) are tightly controlled in non-transformed cycling cells, the latter being deregulated in cancer[5]. Centrioles duplicate in S phase, with the formation of a new centriole next to each pre-existing one, that subsequently elongates until mitosis[6–8]. The two newly formed centrosomes migrate to opposite poles during mitosis, contributing to bipolar spindle formation and appropriate chromosome segregation.

Centrosomes were identified more than one century ago by Van Beneden[9] and Boveri[10] who first proposed a key role for centrosome amplification (>2 centrosomes per cell) in promoting aneuploidy and tumorigenesis[11]. Accordingly, abnormalities in centrosome structure and number have been detected in various tumours since the nineties and associated with genomic instability and poor prognosis[5,12–15]. However, these small structures remained understudied until the advent of sensitive proteomics and RNAi screens, which identified their components. Manipulation of their expression uncovered novel functions for centrosome amplification in promoting features of tumorigenesis, namely chromosomal instability and invasiveness[16,17]. Moreover, centrosome amplification was recently shown to trigger tumorigenesis in vivo[18]. Finally, while non-transformed cells normally die or stop proliferating after abnormal mitosis due to centrosome amplification, cancer cells use mechanisms to cope with this abnormality[19]. With these findings, centrosome amplification and associated survival mechanisms became appealing targets in cancer therapy. Presently, drugs that either prevent centrosome duplication (i.e. a PLK4 inhibitor[20]) or target centrosome amplification survival mechanisms (i.e. HSET inhibitors[21,22]) are in clinical trials or under development, respectively.

However, the identification of centrosome amplification origins and frequency among and within different tumours is critical for its clinical exploitation. Until now, cell division failure and deregulation of the centrosome duplication machinery are the two main mechanisms known to experimentally induce centrosome amplification[23]. However, their relative contributions are not known in cancer, mostly due to technical challenges of studying such small structures. In addition, the research

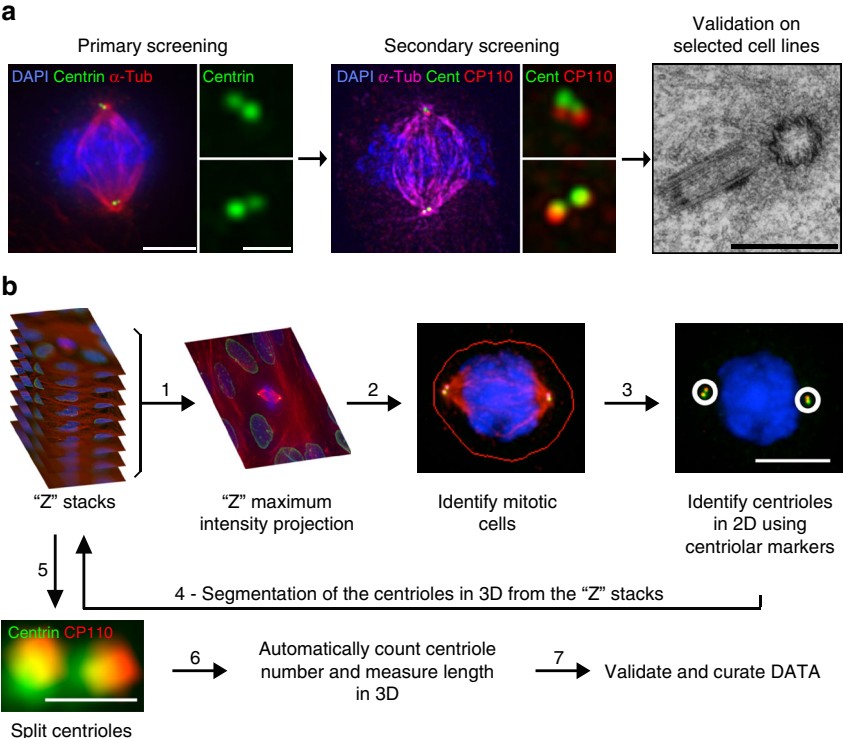

**Fig. 1** Quantifying centriole number and length in cancer cells. **a** Screen overview. Step 1 and 2: primary and secondary immunofluorescence screenings of centrioles in the NCI-60 cancer cell lines using centrin (green) and centrin/CP110 (red) as centriolar markers, respectively. For each screening, we visually selected 50 mitotic cells per each of the cell lines and programmed the microscope to automatically acquire images as Z-stack using a ×100 objective. Cells were also stained with DAPI (blue) and α-tubulin antibody (red-primary screening, magenta-secondary screening). Scale bar 5 µm, insets 1 µm. Since centriole dimensions (250 nm width and 400 nm length) are close to the resolution limit of light microscopy, we selected some cell lines with extreme phenotypes for validation and further investigation, using transmission electron microscopy (TEM), scale bar 500 nm. Please note that all the pictures presented are from a non-cancerous cell line, RPE-1, used as a negative control in this study. **b** Overview of the semi-automatised quantitation of centriole alterations. Briefly, maximum intensity projections, obtained from all the stacks collected per field of view, were created (step 1) and mitotic cells were automatically segmented from a background of interphasic cells using a spatial correlation coefficient based on both DAPI and α-tubulin signals, both of which are brighter in mitotic cells (step 2). Subsequently, centrioles were individually segmented using the centrin staining of each mitotic cell (step 3). In the secondary screening, centrioles were independently identified using both centrin and CP110 staining, and only the structures where these two different markers co-localised were kept for further analysis. Once identified in the 2D projections, all centrioles were segmented in three-dimensions (3D), automatically split into individual centrioles and measured in 3D (steps 4, 5 and 6). The number of centrioles per mitotic cell and their individual lengths were stored, together with a gallery of annotated images (see outputs of steps 2 and 3 for examples). These galleries were verified and manually curated twice (at steps 3 and 5). All remaining steps, from 1 to 6, were automatically performed by the developed algorithm. Scale bar 5 µm, insets 0.5 µm

performed in this area is hindered by: (i) the heterogeneity of methods to study centrosomes, precluding comparisons between studies, (ii) the quantification of centrosome alterations is biased by the limited thickness of paraffin-embedded tissue samples[12]. In view of these limitations, a systematic survey of centriole abnormalities is imperative.

To assess the frequencies of centrosome abnormalities at the single cell level amongst different cancer types, we chose the NCI-60 panel of human cancer cell lines, derived from nine distinct

tissues, as a repository of cancer diversity[24,25]. Importantly, several parameters, critical for a cohesive understanding of the origin and consequences of centrosome abnormalities in cancer, have been characterised in this panel, including: p53, ploidy status and RNA expression[25–30]. Here, we develop a pipeline to semi-automatically measure centriole number and length in mitotic cells. We find that, in addition to centriole amplification, deregulation of centriole length is a recurrent feature of cancer, promoting centriole amplification via both centriole fragmentation

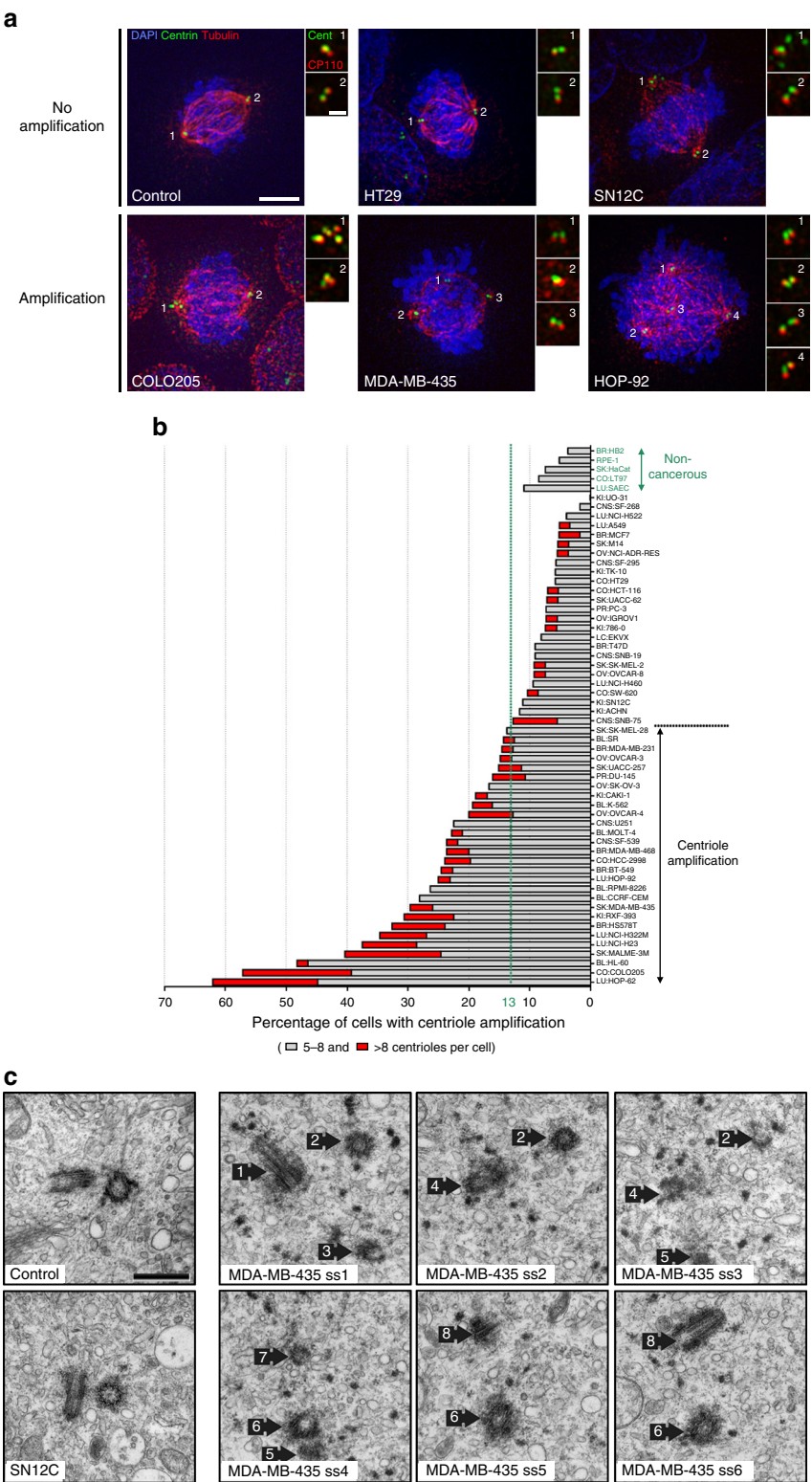

and ectopic procentriole formation. Centriole over-elongation also induces the formation of enlarged centrosomes, with increased MT nucleation capacities, enhancing chromosome missegregation. Altogether, our work establishes centriole amplification and over-elongation as important features of cancer biology, the latter enhancing MT nucleation and chromosomal instability (CIN), two known tumorigenic features. Moreover, our extensive overview of centriole defects in the NCI-60 panel, combined with the publicly available information on its gene expression and drug resistance, will allow further insights on centriole regulation and the development of clinical applications based on centriole aberrations.

## Results

**A semi-automated survey of centriole abnormalities**. To assess the frequencies of centrosome defects in different cancers, we designed a semi-automated and systematic survey to quantify both centriole number and length in the NCI-60 panel of cancer cell lines (Fig. 1a). Given their small size, we developed an algorithm to quantify and measure centrioles in 3D (Fig. 1b and Methods). As both centriole number and length vary throughout the cell cycle, we analysed only mitotic cells, which have a fixed number (4) of fully elongated centrioles.

We performed a primary screening to rapidly estimate the penetrance of centriole amplification and over-elongation. We used centrin to count and measure centrioles as it localises early to the distal end of centrioles, maximising their detection. Furthermore, centrin has been successfully used (i) in centriole-related screens[31], (ii) to track centrioles by live imaging[32], (iii) to score centrosomal defects in cancer[12] and (iv) as a readout for centriole size[33]. Our results show that, while non-transformed cells (human retinal pigmented cells, RPE-1) display no centriole amplification, cancer cell lines have variable degrees of this abnormality (from 5.6 to 90.7% of cells with >4 centrioles, Supplementary Table 2). A total of 14 cell lines exhibit less than 15% of cells with amplification, 17 display between 15 and 30% and 29 have more than 30% of amplification. Centriole size is also controlled and rather homogeneous in non-transformed cells (ref.[34], RPE-1 show no over-elongation), but surprisingly variable in the NCI-60 panel (from 0 to 51.7% of cells with overly long centrioles, i.e. longer than 500 nm-twice the length of a normal-length centriole measured using centrin staining, Supplementary Table 2). A total of 49 cell lines exhibit less than 15% of cells with over-elongation, 7 display between 15 and 30% and 4 have more than 30% of over-elongation.

Although centrin is a widely used marker of all centrioles, false positives may arise as centrin can be present in centriolar satellites, small electron-dense granules[35,36], justifying the need of

a validation screening. For this, we selected the top 50% of cell lines of amplification and over-elongation (35 cell lines) and a subset (18) of the less-defective ones to investigate the presence of false negatives (Supplementary Table 2). We used centrin in combination with a second centriolar marker, CP110, to reliably label centrioles (Supplementary Fig. 1a) and only accounted structures positive for both markers. With this strategy, we identified cells with and without centriole amplification or over-elongation (Figs 2a and 3a). Finally, to define the cut-off and estimate the variability of centriole amplification and over-elongation in non-cancerous cells, we quantified centriole number and length in 5 non-cancerous cell lines from different tissues: RPE-1, HB2, HaCat, LT97 and SAEC (Figs 2b and 3b). On average, 7% ± 3 and 1% ± 2 of cells display centriole amplification and over-elongation, respectively, in these cell lines. Therefore, we set the cut-off for centriole amplification and over-elongation to 13% and 5%, respectively (average + 2 s.d.). The outputs of the secondary screening are depicted in Figs 2b and 3b. Several cell lines included in the top 50% of amplification or over-elongation in the primary screening did not significantly display these anomalies in the secondary screening (Supplementary Table 2). This inconsistency might be due to: (i) lack of sample and/or counts reproducibility, (ii) centrin labelling satellites besides centrioles or (iii) the presence of CP110-negative centrioles. To address this, we compared the automated centrin counts between both screenings in four cell lines displaying high discrepancies between screenings and observed their similarity (Supplementary Fig. 1b). CP110-negative and centrin-positive structures are negative for other centriolar/centrosomal markers in SN12C cell line, which displays the highest discrepancy between screenings (Fig. 2a and Supplementary Fig. 1a). Finally, while validating our screening by TEM (Fig. 3c, d), we observed cells displaying supernumerary centrioles in MDA-MB-435 cell line (positive control for amplification) but never in SN12C (Fig. 2c). These data suggest that centrin labels other structures than centrioles in SN12C cell line, and likely in the other cell lines displaying high discrepancies between the two screenings. Altogether, our observations demonstrate that centrin-positive structures, which are negative for a second centriolar/centrosomal marker, are unlikely bona fide centrioles, validating the results of the secondary screening.

**Centriole number and size are often increased in cancer**. Our secondary screening confirms that non-transformed cells display low levels of centriole amplification (≈7%; Fig. 2b). Our data also shows that approximately half of the cancer cell lines (28) display more than 13% of amplification, threshold established based on our controls, suggesting this abnormality is widespread. The

**Fig. 2** Centriole amplification is widespread in cancer cells. **a** Immunofluorescence images of cell lines without (upper panel) and with (lower panel) centriole amplification. Cells were stained with DAPI (blue), centrin (green), α-tubulin (red) and CP110 antibodies (red-insets). Scale bar 5 μm, insets 1 μm. Note the presence of centrin dots lacking CP110 staining in the SN12C cell line (upper panel-inset 1). **b** Output of the secondary immunofluorescence screenings for centriole number in the NCI-60 panel. To validate the primary screening, the top 50% of amplification and a subset of less-defective ones from the primary screening were incorporated in the secondary screening (see text). The results of the secondary screening are depicted in the bar graph in which cell lines were ranked according to their percentage of mitotic cells with more than four centrioles. To define the cut-off for centriole amplification and the variability of centriole number in non-cancerous cell lines, we quantified centriole number in five non-cancerous cell lines (depicted in green): RPE-1 (retinal pigmented epithelial cells), HB2 (mammary luminal epithelial cells), HaCat (keratinocytes), LT97 (colon adenoma cells) and SAEC (small airway epithelial cells). The average percentage of cells with centriole amplification in the non-cancerous cell lines is 7 ± 3%, therefore we set the cut-off for centriole amplification to 13% (average + 2 standard deviations). 28 cell lines from the NCI-60 panel displayed significant centriole amplification. Note that for most of the cell lines, the majority of cells with amplification showed 5 to 8 centrioles per cell (grey). Cells with more than eight centrioles per cell (red) were less commonly observed. A total of 50 to 60 mitotic cells were analysed per cell line. BR breast, CNS central nervous system, CO colon, BL blood, LU lung, PR prostate, KI kidney, OV ovaries and SK skin. **c** TEM images of control (RPE-1) and SN12C cells with normal number of centrioles (2), and a MDA-MB-435 cell with supernumerary centrioles (8). Each TEM picture represents an individual cell for the control and SN12C cell lines, whereas the remaining pictures are serial sections of the same MDA-MB-435 cell. Scale bar 500 nm

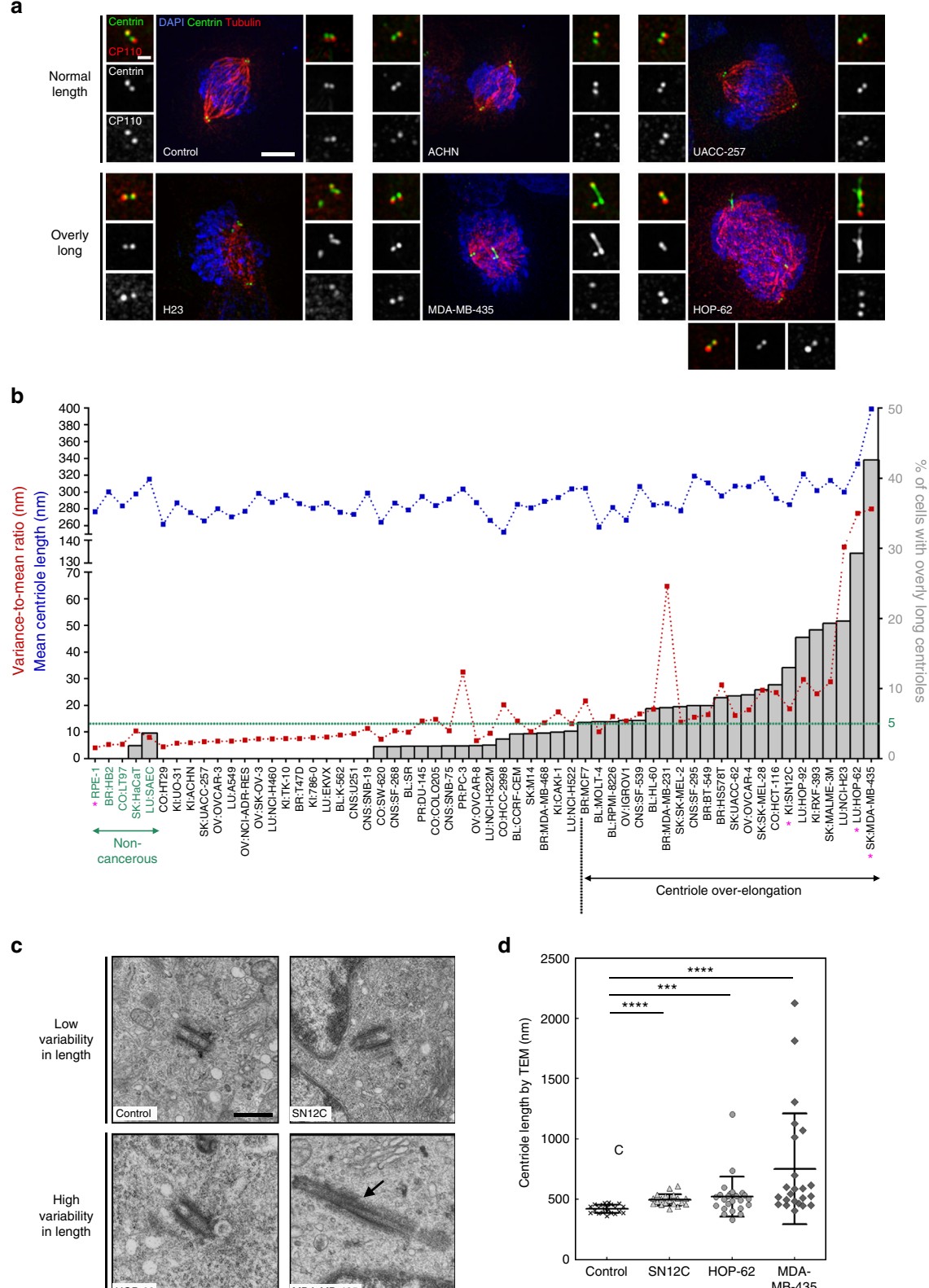

remaining cell lines display lower levels of centriole amplification, similarly to non-cancerous cells (Fig. 2b). Centriole amplification is always present in at least one but never in all cell lines originating from the same tissue, suggesting this feature is tumour-specific rather than tissue-specific (Supplementary Table 2 and Supplementary Fig. 2).

In addition, we observed that 22 cell lines display significant centriole over-elongation compared to non-cancerous cell lines (Fig. 3b and Supplementary Table 2). Cell lines with higher levels of centriole over-elongation belong to breast, skin and lung cancer (Supplementary Fig. 3).

Our results suggest that increase and variability in centriole length are novel recurrent features in cancer. Given the presence of several long centrin-positive structures in some cancer cell lines, we wondered whether these threads could represent abnormal centrin-positive cilia rather than bona fide centrioles. Nearly all centrin-positive structures are negative for ARL13B (a cilia marker[37]) and positive for acetylated tubulin (a stable MTs marker), therefore confirming the centriole nature of centrin threads (Supplementary Fig. 4). For a final validation, centriole length was measured by TEM in 4 cell lines: the non-cancerous RPE-1 cells, and three cancer cell lines displaying increased levels of centriole over-elongation: SN12C, HOP-62 and MDA-MB-435 (Fig. 3b). While centriole length did not exceed 500 nm in RPE-1 cells, we observed slightly longer centrioles in SN12C, long and very long ones in HOP-62 and MDA-MB-435, respectively (Fig. 3c, d). As expected, centrioles appear longer by TEM (e.g. mean of 421 nm in RPE-1) than by immunofluorescence (IF, mean of 277 nm) since centrin is restricted to the medium-distal part of centrioles. Nevertheless, the TEM results mimic the IF data as both show a significant increase in centriole length in the 3 cancer cell lines compared to the control, with SN12C and MDA-MB-435, the least and the most deregulated ones. Our TEM data further corroborates the IF results as centriole length appears homogeneous in RPE-1 (VMR = 2) and SN12C (VMR = 4.5) cells but very heterogeneous in the two cell lines displaying very long centrioles, HOP-62 (VMR = 51.9) and MDA-MB-435 (VMR = 280.3; Fig. 3d). Altogether, the TEM data validates our IF survey of centriole length which established increase and variability in centriole length as novel and recurrent features of cancer cells.

The variability in centriole length was often higher in cell lines with overly long centrioles, (Fig. 3b). This can both reflect inter-cellular (only a subpopulation of cells is affected by centriole over-elongation) and/or intra-cellular variabilities (only one centriole in each cell is affected). To address this, we analysed the distribution of the number of overly long centrioles per cell in the NCI-60 secondary screening. Cells displaying centriole over-elongation have mostly only one (87%), sometimes two (11%) and rarely more than two (2%) elongated centrioles, feature further validated in MDA-MB-435 cells (Supplementary Fig. 5a, b). This intra-cellular heterogeneity might reflect that only certain centrioles are permissive to centriole over-elongation. Each cycling cell in mitosis has three generations of centrioles: the grandmother (the eldest), the mother, generated in the previous cycle, and finally, the daughters, born in the present cycle[38]. We investigated if all generations of centrioles were equally affected by centriole over-elongation using a marker of all centrioles (acetylated tubulin) and a daughter-specific marker (STIL). In

MDA-MB-435 cell line, centriole over-elongation rather affects mother/grandmother (98%) than daughter centrioles (Supplementary Fig. 5c, d). Further staining, using the appendages (structure of grandmother centrioles) protein CEP164, suggests that centriole over-elongation likely affects mainly grandmother centrioles in MDA-MB-435 cell line (Supplementary Fig. 5e).

**High centriole amplification in aggressive breast cancer.** Classification into distinct molecular subtypes based on expression analysis[39,40] is increasingly used to establish prognosis and to predict treatment response in several cancers, e.g. in breast cancer[41]. Interestingly, all cell lines representing the basal breast cancer molecular subtype display significant centriole amplification (MDA-MB-231 (15%), MDA-MB-468 (24%), BT549 (25%) and HS578T (33%)) while luminal cancer cell lines do not (MCF7 (5%) and T47D (9%); Fig. 2b, Supplementary Fig. 2 and Supplementary Table 2).

In addition, centriole amplification is associated with the most common subset of colon carcinoma, CIN (chromosome instability, microsatellites stable), as it was detected in some of these cell lines (HCC-2998 and COLO205), but absent in all MSI-H cell lines (microsatellites instability, hyper-mutated), HCT-15, HCT-116 and KM12 (Supplementary Fig. 2 and Supplementary Table 2). Our results suggest that centriole amplification is more prevalent in specific subtypes of breast and colon cancer, which are both associated with chromosome instability and worse prognosis[42,43].

We validated these findings in human breast using 15 hormone receptor positive (luminal) and 10 hormone receptor negative (basal-like) carcinomas (Fig. 4). Similar to cultured cells, we observed a higher percentage of cells with supernumerary centrioles in basal-like breast carcinomas (22%) than in luminal ones (8%; Fig. 4b, Supplementary Fig. 6 and Supplementary Table 2). Our patient data supports the results of our systematic survey demonstrating that centriole amplification is more frequent in basal-like than in luminal human breast carcinomas suggesting that centriole amplification specifically occurs in more aggressive molecular breast tumour subtypes.

**Centriole amplification, p53 loss and ploidy deregulation.** The strongest molecular association shown with centrosome amplification is loss of p53 function[5,44,45]. To test this, we used the publicly available p53 status of the NCI-60 panel (Supplementary Table 2[29]). p53 is mutated in 80% of the cell lines and most of the cell lines displaying centriole amplification exhibit impairment of p53 function (24/28 cell lines). Nevertheless, this proportion is not statistically different from the one observed in cell lines

**Fig. 3** Increase and variability in centriole length are recurrent in cancer cell lines. **a** Immunofluorescence images of cell lines without (upper panel) and with (lower panel), centriole length deregulation. Cells were stained with DAPI (blue), α-tubulin (red), centrin (green) and CP110 (red-insets) antibodies. Scale bar 5 μm, insets 1 μm. **b** Output of the secondary screening for centriole length in the NCI-60. To validate the primary screening, the top 50% of over-elongation, and a subset of the less-defective ones, were processed in the secondary screening. The results of the secondary screening are depicted in the bar graph (cell lines are ranked according to their percentage of mitotic cells containing at least one overly long centriole). The mean centriole length (blue) and the variance-to-mean ratio (VMR-red), a normalised measure of the dispersion of the distribution, are depicted for each cell line. The cut-off for centriole over-elongation (in green) was set to 5% which corresponds to the average percentage (1%) of cells with centriole over-elongation in the 5 non-cancerous cell lines plus 2 s.d. (2%). 22 cell lines display significant centriole over-elongation. Centriole length was scored in 3D using centrin staining. From 207 to 388 centrioles were measured per cell line. The magenta asterisks label the cell lines that were selected for analysis by TEM. **c** TEM images of longitudinal sections of normal-length centrioles (around 500 nm) in the control (RPE-1) and SN12C cell lines, and of overly long centrioles in HOP-62 and MDA-MB-435 cell lines (740 and 1800 nm, respectively). Note the presence of a crack (arrow) in the overly long centriole of MDA-MB-435 cell. Each TEM picture represents an individual cell. Scale bar 500 nm. **d** Quantification of centriole length, using the Image J software, from TEM images of longitudinal sections of different centrioles for the control (#22, median = 427 nm, VMR = 2.7), SN12C (#20, median = 486 nm, VMR = 4.5), HOP-62 (#25, median = 502 nm, VMR = 51.9) and MDA-MB-435 (#23, median = 546 nm, VMR = 280.3) cell lines. The bar represents the mean ± s.d. ***represents $p < 0.001$ and ****$p < 0.0001$ (Mann–Whitney statistical test)

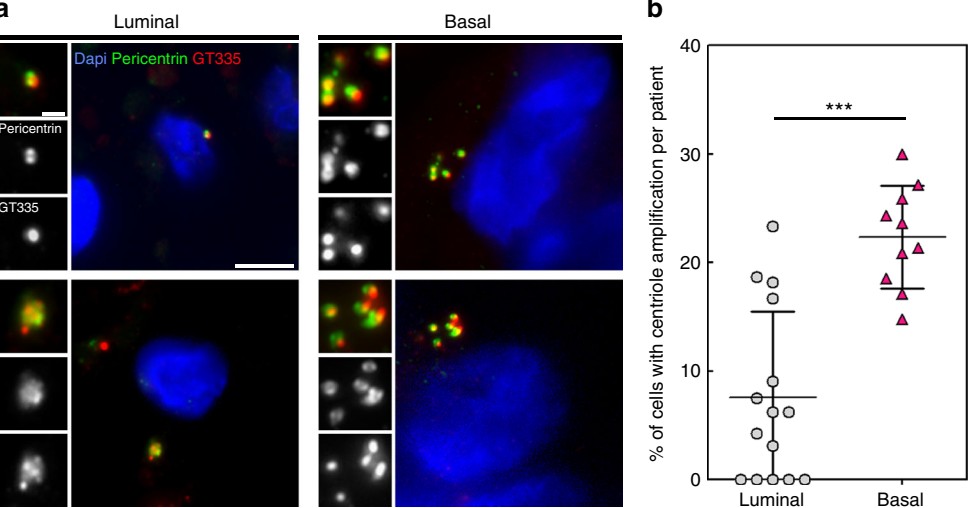

**Fig. 4** Basal-like human breast carcinomas display more centriole amplification than luminal breast carcinomas. **a** Representative immunofluorescence images of sections of human breast tumours showing normal centriole number (2) in interphasic cells of luminal breast carcinoma (upper left panel), and centriole amplification (>4), in both luminal (lower left panel) and basal-like (right panel) breast carcinomas. We labelled centrioles using the centriolar marker GT335, which labels tubulin glutamylation, a modification of tubulin present in centrioles, in co-localisation with the PCM marker, pericentrin. Tissue sections were also stained with DAPI (blue). Please note that we focused our analysis on interphase cells, as it is very difficult to find mitotic figures in breast tumours. Scale bar 5 μm, insets 1 μm. **b** Percentage of cells displaying centriole amplification in each of the 15 patients with luminal breast carcinomas (average 8% ± 2) and in each of the 10 patients with basal-like carcinomas ((on average 22% ± 2), $p = 0.0004$, Mann–Whitney test). Please note that only structures positive for both pericentrin and GT335 were taken into consideration to avoid false positives. All basal-like carcinomas (10 out of 10 patients) exhibited centriole amplification while it was not detected in 5 out of the 15 luminal carcinomas. Between 20 and 133 cells were analysed for each patient (see Supplementary Fig. S6). The bar represents the mean ± s.d

without significant amplification (Supplementary Fig. 7a). Interestingly, almost half of the p53 MT cell lines show no increase in centriole number (18/42, Supplementary Table 2), suggesting that loss of p53 is not sufficient to cause centriole amplification.

Centriole amplification may arise from cytokinesis failure, mitotic slippage and/or cell fusion[23], all of which would also lead to severe ploidy deregulation. Mild ploidy deregulation might also arise from centrosome amplification as the latter induces the formation of lagging chromosomes[17,46]. We wondered whether centrosome amplification correlates with ploidy deregulation, modal chromosome number, numerical chromosome changes and number of structural chromosomal rearrangement[30,47] (Supplementary Table 2). All these parameters define the karyotypic complexity and heterogeneity of each of the NCI-60 cell line and none of them is correlated with centriole amplification (Supplementary Fig. 7b, c). However, 64% of the cell lines with centrosome amplification show ploidy deregulation, suggesting that events affecting both ploidy and centrosome number, such as cytokinesis failure, might induce centriole amplification in some cancers (Supplementary Table 2).

**Centriole over-elongation drives centriole amplification.** As both centriole amplification and over-elongation are widespread in our screen, we investigated their interdependency and observed their correlation (Supplementary Fig. 8). Additionally, the proportion of long centrioles in cells with centriole amplification is statistically different from the expected proportion under the null hypothesis of independence, therefore supporting that centriole over-elongation and amplification are not independent (Fig. 5a). This correlation might reflect that centriole over-elongation triggers amplification, as suggested upon experimentally induced centriole over-elongation in the osteosarcoma cell line, U2OS[48,49]. To further substantiate this direct link in the NCI-60 panel, we overexpressed CPAP, a promoter of centriole

elongation[48], in two cancer cell lines without significant amplification and over-elongation, T47D and SF268. The percentage of mitotic cells with amplification is significantly increased upon CPAP overexpression (from 4% ± 1 to 26% ± 7 for T47D and from 3% ± 1 to 28% ± 7 for SF268; Fig. 5b) therefore confirming that over-elongation triggers amplification.

Centriole over-elongation might induce amplification via ectopic procentriole formation[48,49]. Moreover, it was suggested that overly long centrioles, may fragment[48], which we tested by inducing centriole over-elongation using CPAP overexpression in U2OS cells (Supplementary Fig. 9). While maximum centriole length is achieved at the onset of overexpression, a decrease in length together with an increase in centriole number are then observed (Supplementary Fig. 9a, c). This negative correlation supports that centriole over-elongation induces amplification, perhaps through fragmentation. To confirm this, we prevented centriole biogenesis by inhibiting its master regulator, PLK4, with centrinone B[50], which drastically diminishes centriole number (Supplementary Fig. 9d). Remarkably, CPAP overexpression partially rescues centrinone B treatment with some cells even displaying supernumerary centrioles (Supplementary Fig. 9d). Centrosome amplification via over-elongation is therefore partially independent of centriole biogenesis, but relies partly on fragmentation (Supplementary Fig. 9d), which was further corroborated at the ultra-structural level by TEM (Supplementary Fig. 9e).

We then investigated if centriole fragmentation and ectopic centriole formation occur in the NCI-60 panel using two cell lines displaying high degree of centriole over-elongation, MDA-MB-435 and HOP-62. We examined centriole structure using TEM and Structured Illumination Microscopy (SIM), the latter using acetylated tubulin as a centriole barrel marker. With SIM, we also investigated procentriole nucleation sites using STIL staining. While only normal-length centrioles with a proper barrel shaped structure are visible in non-cancerous cells line (Fig. 5c, upper

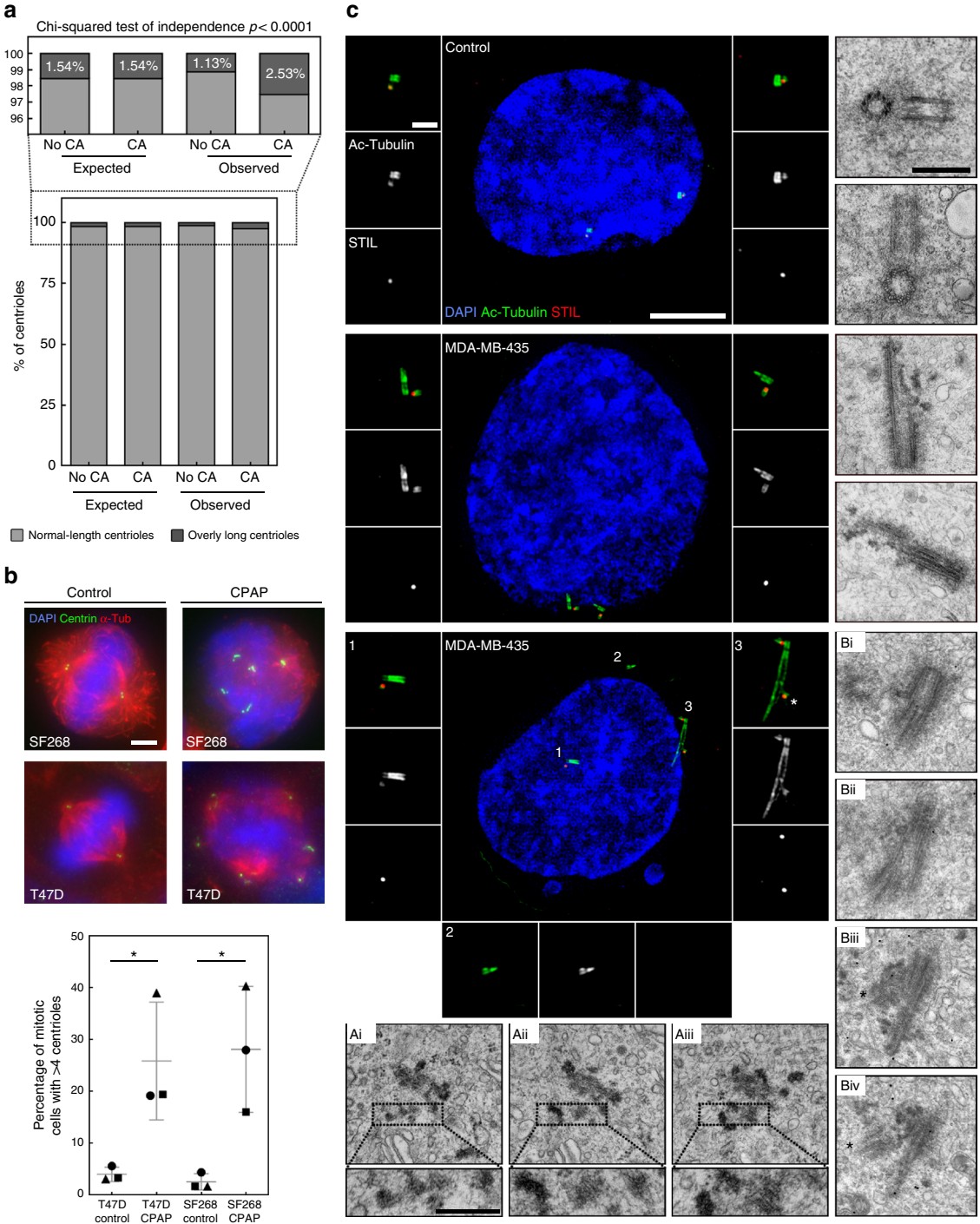

**a** Chi-squared test of independence *p* < 0.0001

Normal-length centrioles
Overly long centrioles

**b** Control / CPAP — SF268, T47D

**c** Control — DAPI Ac-Tubulin STIL — MDA-MB-435

panel), we observed asymmetrically elongated centrioles in MDA-MB-435 cells, (Fig. 5c, middle panel), which often contain breaks (see Fig. 5c, middle panel, lower TEM picture). This result suggests either that only some of the MTs from the barrel have been extended or that overly long centrioles are poorly stable and break. Branched and opened centrioles at the tips were also visualised by SIM in MDA-MB-435 and HOP-62 cells (Supplementary Fig. S10a, b, panel (ii)). These data show that overly long centrioles often exhibit abnormal, and likely unstable, structures at their distal ends. Interestingly, we found evidences of centriole fragmentation by SIM in MDA-MB-435 (fragment which were neither procentrioles (STIL-negative) nor mother centrioles (not associated to a nascent procentriole, unlike other centrioles in the

same cell; Fig. 5c, lower panel, insets 2). Centriole fragmentation was also observed in HOP-62 cells (fragments emanating from long centrioles; Supplementary Fig. 10b, panel (iii)). This was further confirmed by TEM in MDA-MB-435 cells with the presence of centriole fragments surrounding a centriole with a markedly abnormal tip, (Fig. 5c, dotted boxes of Ai, Aii and Aiii).

Induced centriole over-elongation triggers amplification via ectopic procentrioles formation[48,49]. Normally, only one procentriole forms close to each mother centriole proximal-end in control cells (STIL-positive centriole, Fig. 5c, upper panel). Strikingly, we observed that 4% of the overly long centrioles of MDA-MB-435 cells nucleate multiple procentrioles along their length (asterisk in inset 3 of SIM picture of Fig. 5c and

Supplementary Fig. 10c), a phenomenon further confirmed by TEM (asterisk in the EM pictures Biii and Biv of Fig. 5c). Our study discovered two novel sources of centriole amplification in some cancer cells: centriole fragmentation and ectopic procentriole formation upon severe centriole over-elongation.

**Overly long centrioles form over-active centrosomes.** Our survey unravelled centriole length deregulation as a novel common feature of cancer cells. Interestingly centriole length was suggested to set centrosome size[48,51], therefore overly long centrioles should form larger, and over-active, MTOCs. To test this, we first determined if overly long centrioles recruit more PCM than normal-length centrioles. To investigate specifically the effect of "centriole length" on PCM recruitment, we quantified the PCM content (γ-tubulin and pericentrin) of centrosomes with an asymmetric centriole length content in MDA-MB-435 mitotic cells and in mitoses with only normal-length centrioles (thereafter called symmetric mitoses) in RPE-1 and MDA-MB-435 cells (Methods section). Mitoses containing overly long centrioles clearly display an asymmetry in PCM content with poles with longer centriole(s) recruiting more than double PCM amount than the other poles. Contrarily, in symmetric mitoses, almost no difference in PCM content was observed between poles (Fig. 6a and Supplementary Fig. 11a). These results suggest that overly long centrioles recruit more PCM than normal-length centrioles which was further investigated by inferring the interdependency of PCM content and centriole length (centrin intensity) in MDA-MB-435 and HOP-62 cells. Both parameters indeed correlate in both cell lines, confirming that elongated centrioles recruit more PCM (Supplementary Fig. 11b, c).

To further substantiate the direct link between centriole over-elongation and enhanced PCM recruitment, we induced centriole over-elongation in T47D and SF268 and quantified PCM content of normal-length and overly long centrioles (YFP and CPAP overexpression, respectively). Elongated centrioles contain more PCM than normal-length centrioles (Fig. 6b), therefore confirming that overly long centrioles form larger centrosomes which are likely over-active. To confirm this, we analysed the MT regrowth capacities of centrosomes in (i) MDA-MB-435 mitoses displaying overly long centrioles, and (ii) RPE-1 and MDA-MB-435 symmetric mitoses (Methods section). Almost no difference in α-tubulin content was observed between poles in symmetric mitoses (Fig. 6c). Remarkably, in spindles with asymmetric poles, the pole containing longer centriole(s) contains, ≈1,6 times more MTs than the other pole (Fig. 6c and Supplementary Fig. 11d), demonstrating that centrosomes containing overly long centrioles are indeed over-active.

**Elongated centrioles enhance chromosome segregation defects.** Induced centriole over-elongation was suggested to trigger defective cell division, mostly through multipolar mitosis formation due to the accumulation of supernumerary MTOCs[48]. We reasoned that centriole over-elongation could intrinsically, and independently of centriole amplification, be detrimental for mitotic cells as overly long centrioles form over-active centrosomes that might generate unbalanced forces on chromosomes, leading to chromosome instability and aneuploidy, two known features of cancer[52]. To test this, we compared the occurrence of chromosome segregation defects in mitotic MDA-MB-435 cells containing only two centrosomes, with or without asymmetric centriolar content. The incidence of chromosome segregation defects, especially DNA/chromosome bridges, was higher in asymmetric anaphases and telophases (32%), compared to symmetric ones (18%; Fig. 6d and Supplementary Fig. 12a, b).

To further link centriole over-elongation to chromosome segregation defects, we induced over-elongation in T47D and SF268. The proportion of multipolar prometaphases and metaphases, a known source of chromosome instability[17,46], statistically increases upon CPAP overexpression (from 14% ± 1 to 34% ± 8 in T47D and from 3% ± 0 to 18% ± 4 in SF268; Fig. 6e). Chromosome missegregation during anaphases and telophases is also significantly enhanced upon CPAP overexpression (from 28% ± 0 to 44% ± 4 in T47D and from 17% ± 4 to 40% ± 4 in SF268, Fig. 6e and Supplementary Fig. 12c). Our results show that induced centriole over-elongation enhances multipolar mitosis formation and chromosome segregation defects in cancer cell lines.

In conclusion, our results suggest that overly long centrioles induce the formation of over-active centrosomes which enhance chromosome missegregation, both directly and indirectly via centrosome amplification. This likely leads to aneuploidy and therefore may participate in tumorigenesis (Fig. 7).

## Discussion

Recent work showed the importance of centriole amplification in promoting tumorigenesis[18] and as a target for cancer therapy[20,21], raising the need to determine its prevalence and origins in cancer. Here, we provided a sensitive and robust systematic survey of centriole abnormalities in cancer cells. Our survey established that centriole over-elongation and amplification are widespread in cancer, the latter correlating with aggressiveness in breast and colon cancer cell lines. Our follow up studies showed that overly long centrioles form over-active centrosomes that enhance chromosome missegregation. Furthermore, we provide the first explanation for the occurrence of centrosome amplification in cancer as we demonstrate that centriole over-elongation generates

**Fig. 5** Centriole over-elongation drives centriole amplification in cancer cells via ectopic procentriole formation and centriole fragmentation. **a** Centriole elongation and amplification are not independent. Higher proportion of overly long centrioles (>500 nm) in cells with centriole amplification (CA), (*p* < 0.0001, Pearson's Chi-squared test), compared with the expected proportions under the null hypothesis of independence. The expected and observed percentages of overly long centrioles in cells with no CA and CA are shown, together with a detailed view above, given the low frequency of overly long centrioles. **b** Induced centriole elongation triggers centriole amplification in cancer cell lines. CPAP was transiently overexpressed for 96 h in two cell lines from the NCI-60 panel, T47D and SF268, which do not normally display centriole over-elongation. Cells were stained with DAPI (blue), alpha-tubulin (a marker of MTs) and centrin (used as a readout for centriole number) antibodies (Scale bar 5 μm). 3 independent experiments performed (depicted with squares, triangles and circles). 80–140 cells were counted per condition and per experiment. The bars represent the means ± s.d. *Represents a *p* < 0.05 (one-tailed unpaired *t* test with Welch's correction). **c** Centriole fragmentation and ectopic centriole formation in cancer cells with overly long centrioles. Examples of structured Illumination Microscopy (SIM) and TEM pictures showing normal-length centrioles in the control cell line (upper panel), overly long centrioles with defective structures (middle panel), centriolar fragments (lower panel, inset 2 for SIM and insets of Ai, ii, iii for TEM) and ectopic procentrioles along the overly long centrioles (lower panel, labelled by asterisks in inset 3 for SIM and Biii and iv for TEM) in the MDA-MB-435 cell line. For SIM, cells were stained with DAPI (blue), acetylated tubulin (green) and STIL (red) antibodies, and were subjected to a 2 h cold treatment, prior to fixation, to depolymerise the cytoplasmic MTs. For TEM, each picture represents an individual cell except for Ai, ii, iii and Bi, ii, iii and iv, where the pictures are serial sections of the same cell. Scale bar: 5 μm (SIM) or 500 nm (TEM). Insets: 1 μm (SIM) or 250 nm (TEM)

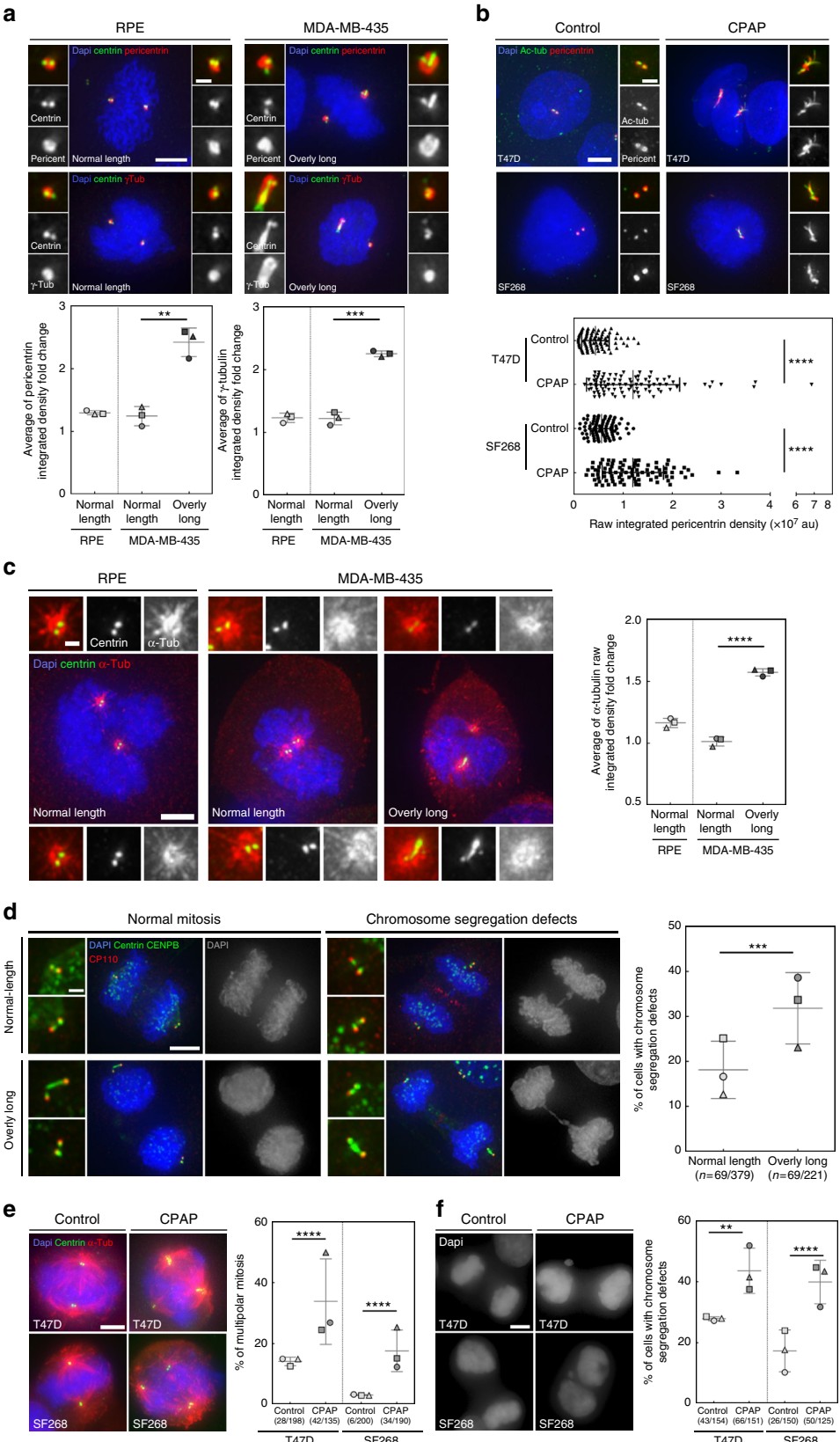

supernumerary centrioles, via both centriole fragmentation and ectopic procentriole nucleation. Our work represents an important asset to uncover the origins and consequences of centrosome defects in cancer, and in the development of new tools for cancer diagnosis and therapeutics.

The NCI-60 panel encompasses 60 cancer cell lines from nine distinct tissues, and has been widely used as a repository of cancer diversity, since the majority of the NCI-60 cell lines genetically represent their corresponding tumour types[24]. Our survey reinforces the use of this panel as we observed a strong correlation

between the findings in breast cancer cell lines and patient samples.

Our survey demonstrated that half of the NCI-60 cell lines, and all analysed tissues, show significant centriole amplification compared to non-cancerous cell lines, confirming how widespread this phenomenon is in cancer. We also identified increase and variability in centriole length as common features of cancer cells, explaining previous observations of overly large and/or long centrin-positive structures in both breast cancer and plasma cell neoplasms in situ[53,54].

Our survey unveiled discrepancies in the penetrance of centriole amplification within different cell lines from the same tissue, indicating that centriole amplification is tumour-specific rather than tissue-specific. We showed that centriole amplification is more prevalent in specific molecular subtypes of breast (basal-like) and colon (CIN molecular subtype) cancer cell lines which represent particularly aggressive carcinomas, associated with poor prognosis[42,43]. Specific molecular features preferentially present in these carcinomas may underlie the presence of high levels of centrosome amplification. For instance, basal-like breast carcinomas have an increased incidence in patients with germline *BRCA1* mutations[55], shown to induce centrosome overduplication[56]. In CIN colon cancer, centrosome amplification may arise through Aurora-A gene amplification, whose overexpression leads to supernumerary centrosomes[45,57,58]. Our data confirm that centrosome amplification is associated with malignant features, endorsing its potential as a tumour classifier to establish prognosis and predict treatment response. The fact that centrosome alterations are quite penetrant in cancer and rare in non-cancerous cell lines, supports its use as a target in aggressive tumour therapy[20,23].

The large variability in the percentage of cells with supernumerary centrioles observed between cell lines confirms the concept of an intrinsic centrosome amplification "set point" for each cell line, proposed by Wong et al.[50]. By reversibly inhibiting centriole biogenesis, these authors generated dividing cells without centrosomes in different cancer cell lines. After inhibitor washout, cells form massive amounts of centrosomes de novo, but gradually recover their initial level of centrosome amplification[50]. This may reflect a tumour-specific dynamic equilibrium between stochastic emergence and death of cells with supernumerary centrioles. Accordingly, a recent study followed cell fate after induced centriole overduplication and highlighted that their offspring dies during the next cell cycle[59]. In conclusion, while cells with multiple centrosomes may arise and be present in the population, part of their progeny may die. Cells with centriole

amplification may survive only if beneficial to the overall population (e.g. by promoting invasiveness[16] and/or yet unknown non-cell autonomous effects that promote survival). Further understanding of the "centrosome set point" is critical to successfully use centrosome amplification as a target for cancer therapy.

Previously, both the limited use of centriole markers and absence of genetic characterisation of tumours precluded proper and systematic understanding of the origin of centriole abnormalities[5,12]. Our screening provided an opportunity to address this important question. We focused initially in understanding the role of p53 and ploidy deregulation, given existing controversy on their importance[23,60,61]. While most cell lines with amplification lost p53 function, this event is not sufficient to trigger this abnormality, therefore reinforcing the idea that loss of p53 function is a prerequisite to sustain centrosome amplification across cancer[44,45,62], rather than a direct cause. Finally, we observed that only a subpopulation of the cell lines displaying amplification has significant increase in ploidy, suggesting different origins of ploidy and centrosome deregulations. Given that cells were in a steady state, we cannot exclude that a common event affects both ploidy and centrosome homoeostasis, as this might be hidden by differential evolution of ploidy and centrosome number. Further studies looking at the evolution of populations after triggering ploidy and centrosome number deregulation are needed to understand how these processes are related in cancer.

Our study identified a novel cause of centriole amplification in cancer, centriole over-elongation. This defect affects only a subpopulation of cells, therefore reflecting inter-cellular heterogeneity that can reflect a dynamic equilibrium between genesis and death of cells with centriole over-elongation. In addition, we observed intra-cellular heterogeneity with usually only one or two overly long centrioles per mitotic cell. This might reflect that centriole length regulators are rate-limiting or alternatively that not all centrioles are equally permissive to centriole length deregulation. We rather substantiated the latter hypothesis by showing that centriole over-elongation mostly affects grandmother centrioles in MDA-MB-435 cell line. Additional studies are now required to further investigate those hypotheses.

We showed that overly long centrioles can fragment and ectopically form procentrioles in cancer cell lines. These phenomena likely explain previous observations of long centrin fibres and electron-dense fragment-like microtubule complexes in breast carcinoma[53]. This phenotype could result from the deregulation of the expression of centriole length

**Fig. 6** Centriole over-elongation leads to the formation of over-active centrosomes. **a, b** Overly long centrioles recruit more pericentriolar material than normal-length centrioles. **a** RPE-1 (control, only normal-length centrioles) and MDA-MB-435 (with normal-length and overly long centrioles) cells were stained with DAPI (blue), centrin (green), and γ-tubulin or pericentrin, (PCM components, red) antibodies (for details see Supplementary Fig. 11a and methods). 3 independent experiments performed (squares, triangles, circles). A total of 37–62 centrosomes quantified per condition and per experiment. One-tailed unpaired t test with Welch's correction. **b** Induced centriole elongation triggers enhanced PCM recruitment. CPAP was overexpressed for 48 h in two NCI-60 cell lines, T47D and SF268, which do not display overly long centrioles. Cells were stained with DAPI (blue), pericentrin (red) and acetylated tubulin (green) antibodies. Compilation of three experiments (around 30 centrosomes accounted per condition and per experiment). Mann–Whitney test. **c** Overly long centrioles nucleate more MTs than normal-length centrioles. Microtubule regrowth assay in RPE-1 and MDA-MB-435 cell line. Cells were stained with DAPI (blue), α-tubulin (red) and centrin (green) antibodies (see Supplementary Fig. 11d and methods). Three independent experiments performed (squares, triangles, circles). A total of 30–82 centrosomes accounted per condition and per experiment. One-tailed unpaired t test with Welch's correction. **d** Chromosome segregation defects (see Supplementary Fig. 12a, b) are increased in mitotic cells with overly long centrioles. MDA-MB-435 cells were stained with DAPI (blue), CENPB (centromere, green), centrin (green) and CP110 (red, CP110 antibody sometimes labels the midbody) antibodies. (200 cells counted per experiment, n = 3 squares, triangles, circles). z score test. **e, f** Induced centriole elongation enhances multipolar mitosis formation and chromosome segregation defects (for details see Supplementary Fig. 12). CPAP was transiently overexpressed for 96 h in T47D and SF268 cell lines. Cells were stained with DAPI (blue), α-tubulin (red) and centrin (green) antibodies. Three independent experiments performed (squares, triangles, circles). Between 30–87 mitosis for **e** and 38–55 for **f** accounted per condition and per experiment. z score test, one-tailed. For all images: scale bar: 5 µm, insets: 1 µm (except for **b**, insets: 2 µm). For all graphs, the bars represent the means ± s.d. For all tests, **$p \leq 0.01$ and ***$p \leq 0.001$ and ****$p \leq 0.0001$

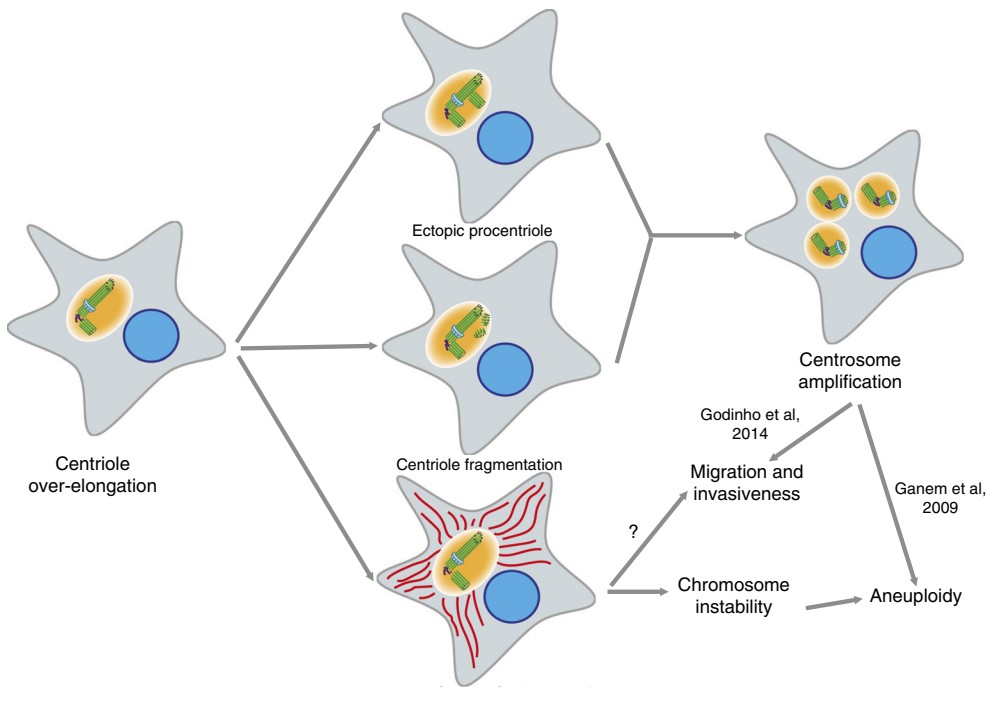

**Fig. 7** Cellular consequences of centriole length deregulation. In cancer cells, over elongated centrioles induce centriole amplification through centriole fragmentation and/or ectopic procentriole formation along the elongated centrioles. Elongated centrioles also generate larger MTOCs with a higher capacity to nucleate MTs that enhance chromosome instability during mitosis. Both scenarios could give rise to aneuploidy and might as well induce invasiveness, therefore centriole length deregulation might participate to tumour initiation and progression

regulators[48,49,63], or loss of cell cycle control[64]. Further investigations including mining the publicly available NCI-60 omics data are now required to identify the causes of centriole length deregulation in cell lines displaying over-elongation. This strategy is critical to exploit this defect in cancer biology, and will likely provide a better understanding of the mechanisms underlying centriole length regulation in normal cells.

Supernumerary centrosomes induce aneuploidy, via merotelic attachment of chromosomes during mitosis, as well as invasiveness, due to enhanced MT nucleation capacity resulting in extracellular matrix degradation and RAC1 activation[16,17]. We propose that overly long centrioles may induce these features via the formation of supernumerary centrioles. In addition, overly long centrioles might directly trigger invasiveness as they form over-active centrosomes which enhance MTs nucleation and chromosome missegregation. We indeed observed an increase in the occurrence of chromosome misalignments, polar asymmetry and lagging chromosomes upon centriole over-elongation. These defects likely result from the formation of asymmetric mitoses with the pole containing longer centrioles, being over-active, capturing more chromosomes than the other pole. Similarly, mitoses with uneven numbers of daughter centrioles between the two poles were recently shown to display asymmetric PCM content and, therefore MT nucleation capacities. This asymmetry in MT content leads to unequal kinetochore capture therefore increasing the rate of chromosome missegregation[59]. Surprisingly, we also observed an increase in the occurrence of chromosome bridges upon centriole elongation. Overly long centrioles may induce this defect by triggering mitotic delays, therefore inducing Aurora B kinase-dependent telomere uncapping and chromosome fusion[65]. Alternatively, chromosome bridges might be a consequence of the formation of lagging chromosomes. The latter can generate micronuclei with fragmented chromosomes, which in the next cell cycle cause bridges[66,67]. Further

investigations are now required to decipher how centriole elongation triggers chromosome segregation defects. Altogether, our work demonstrates that centriole over-elongation enhances microtubule nucleation and chromosomal instability, two known tumorigenic features. Further studies are now required to determine if centriole over-elongation is sufficient to trigger tumorigenesis in vivo.

Our study provides a thorough and comprehensive compendium of centriole number and length abnormalities in the NCI-60 panel of cancer cell lines. Our work supports that centrosome amplification is widespread in cancer and uncovers centriole length deregulation as a common feature in cancer cells. Because these changes are specific to cancer, these findings unravel novel and promising opportunities in the fields of cancer diagnostic and therapy. Our survey combined with the publicly available data of the NCI-60 cell lines are powerful and invaluable tools for the identification of the molecular mechanisms by which cancer cells lose precise control of centriole number and length. By making our data publicly available, we will substantially expand the characterisation of the NCI-60 panel of cancer cell lines and consequently its use by the scientific community in the pursuit of novel clinical and diagnostic tools in cancer research.

## Methods

**Cell culture**. All NCI-60 cell lines were cultured in their respective media (see Supplementary Table 1), which was supplemented with fetal bovine serum (FBS, Gibco), Penicillin (100 IU mL$^{-1}$, Gibco) and Streptomycin (100 µg mL$^{-1}$, Gibco). Note that the origins of all the NCI-60 cell lines are stated in Supplementary Table 1. NCI/ADR-RES, MDA-MB-435 and SNB-19 were reported to be misidentified cell lines. To keep the NCI-60 panel entire, and be able to compare with other studies using this panel, we kept these cell lines but affiliated them to their relative correct tissue of origin. All NCI-60 cells used in the secondary screening were tested for mycoplasma, the remaining cell lines were not. None of the cell lines used in this study were authenticated. hTERT-RPE-1 (kind gift from Lars Jansen—IGC) and HB2 (kind gift from Fanni Gergely—Cancer Research UK) and HaCaT (kind gift from John Marshall—BCI-QMUL) cell lines were culture in

DMEM (Gibco) media with 10%FBS. For HB2 cell line, the media was supplemented with 5 µg mL$^{-1}$ of hydrocortisone and 10 µg mL$^{-1}$ of insulin. The LT97 (kind gift from Sérgia Velho—I3S) cell line was cultured in a 4/1 mixture of HamF12 (Gibco, supplemented with 10 mM Hepes, Calbiochem) and L15 (Gibco) media supplemented with 2% FBS, 10 µg mL$^{-1}$ insulin, $2 \times 10^{-10}$ M triiodotyronin (Sigma), 2 µg mL$^{-1}$ transferrin (Gibco), 1 µg mL$^{-1}$ hydrocortisone and 30 ng mL$^{-1}$ epidermal growth factor (Sigma). The SAEC (kind gift from Tyson Sharp—BCI-QMUL) cell line was cultured in SAGM$^{TM}$ BulletKit$^{TM}$ media (Lonza). The U2OS cell line (kind gift from Pierre Gönczy—EPFL) was cultured in Mc Coy's media (Gibco). All media were supplemented Penicillin (100 IU mL$^{-1}$, Gibco) and Streptomycin (100 µg mL$^{-1}$, Gibco). L-Glutamine (2 mM, Gibco) was also added to all glutamine-deprived media. All cell lines were grown in a humidified 5% CO$_2$ incubator at 37 °C.

**Human breast carcinoma samples.** Formalin-fixed and paraffin-embedded human breast tumour samples were consecutively retrieved from the files of the Department of Pathology, Hospital Xeral-Cies, Vigo, Spain. This series comprises 15 luminal carcinomas and 10 basal-like tumours. The status of the oestrogen receptor (ER), progesterone receptor (PR), epidermal growth factor receptor 2 (HER2), antigen Ki67, and the basal markers epidermal growth factor receptor, cytokeratin 5, cytokeratin 14, P-cadherin and Vimentin was previously characterised for all of tumour cases. According to their immunoprofile, breast tumour samples were classified as luminal (ER+, PR+, HER2− and Ki67−) or basal-like carcinomas (ER−, PR−, HER2−, basal marker+). Representative tumour areas were carefully selected and at least two tissue cores (0.6 mm in diameter) were deposited into a tissue microarray. This study was conducted under the national regulative law for the handling of biological specimens from tumour banks, with samples being exclusively used for research purposes in retrospective studies. Informed consent was obtained from all human participants.

**Immunofluorescence staining.** Adherent cell lines were directly grown on glass coverslips. Suspension cell lines were pelleted and then re-suspended in 100 µL of 1× Dulbecco's phosphate buffered saline (PBS) solution without Calcium and Magnesium (Biowest) and cytospinned onto slides using a Wescor Inc 7620 Cytopro™ Cytocentrifuge (500 rpm for 5 min at medium acceleration). Subsequently, both adherent and suspension cells were fixed using cold methanol for 10 min at −20 °C and, afterwards, incubated with 1× PBS containing 10% FBS for 30 min at room temperature. Immediately after, cells were incubated for 1 h and 30 min at room temperature with primary antibodies diluted in 1× PBS + 10% FBS. After the incubation with the primary antibody, cells were washed three times, 5 min each, using a 1× PBS solution. After the washes, the cells were incubated for 1 h at room temperature with secondary antibodies (Alexa 488 or Fitc, Rhodamine, Alexa 647 or Cy5, Jackson), diluted at 1/500 in 1× PBS with 10% FBS. Immediately after, cells were washed three times, 5 min each, in 1× PBS solution, where in DAPI was added to the second wash to stain DNA. The coverslips obtained by this procedure were mounted on slides using ProLong® Gold Antifade Reagent (Molecular Probes®). The slides were kept 24 h at room temperature to allow the mounting media to cure. For Human breast carcinomas, 3 µm-thick tissue sections were deparaffinised in Clear-Rite-3 (Thermo Scientific, USA, CA) and rehydrated using a series of solutions with decreasing concentrations of ethanol. High temperature (98 °C, 60 min) antigenic retrieval with Tris-EDTA pH = 9.0 (LeicaBiosystems, UK) was performed, followed by incubation with UltraVision protein block (Thermo Scientific) for 30 min at room temperature. The slides were, afterwards, incubated with mouse anti-GT335 (1/800 dilution, Adipogen Ref. AG-20B-0020-C100) and rabbit anti-pericentrin (1/250 dilution, Abcam AB4448) in UltraAb diluent (Thermo Scientific) overnight at 4 °C. The sections were then washed three times, 5 min per wash, with 1× PBS + 0.02% Tween20 before a 1 h room temperature incubation with the secondary antibodies, anti-IgG rabbit coupled to Alexa 488 and anti-IgG mouse coupled to Alexa-594 (Invitrogen), diluted at 1/500 in PBS. Finally, sections were washed extensively with 1× PBS + 0.02% Tween20 and then counterstained and mounted with Vectashield containing DAPI (VectorLabs, CA, USA).

Imaging was performed on a Zeiss Imager Z1 inverted microscope, equipped with an AxioCam MRm camera (Zeiss) and ApoTome (Zeiss), using the ×100 1.4 NA Oil immersion objective. Images were taken as Z-stacks in a range of 10–14 µm, with a distance between planes of 0.3 µm, and were deconvolved with AxioVision 4.8.1 software (Zeiss). Only the structures positive for GT335 (centriolar marker) and pericentrin (PCM marker) were analysed and scored. Countings of centrioles were performed in breast cancer cases, with a minimum of 20, and a maximum of 133 cells scored per case. Statistical analyses were conducted using IBM SPSS Statistics (Version 22.0. Armonk, NY, IBM Corp). The statistical significance of the associations between categorical variables was determined using the $\chi^2$ test. A Mann–Whitney statistical test was used to compare centrioles scoring between luminal and basal-like carcinomas. A two-tailed 5% significance level was considered as statistically significant ($p < 0.05$).

**Primary immunofluorescence screening.** Cells were stained with DAPI, centrin-2 (1/100 dilution, Ref. sc-27793-R, Santa Cruz Biotechnology,) and alpha-tubulin (1/100 dilution, Ref. DM1A, Sigma) antibodies. Images were acquired on an Applied

Precision DeltavisionCORE system, mounted on an Olympus IX71 inverted microscope, using a ×100 1.4 NA Oil immersion Olympus UPlan SAPO objective. Data were acquired with a Photometrics Cascade II EMCCD camera (final pixel size: 0.128 µm) through Applied Precision SoftWorx v5.5.1 Resolve3D acquistion software using DAPI (Ex: 390/18; Em: 457/50), FITC (Ex: 475/28; Em: 528/38), TRITC (Ex: 542/28; Em: 617/73) and CY5 (EX: 632/22; Em: 685/40) bandpass filters for DAPI, Ax488, Rhodamine and Ax647 fluorophores, respectively, with a Lumencor Spectra 7 SSL light engine. Images were taken as Z-stacks in a range of 10–14 µm and were later deconvolved with Applied Precision's SoftWorx deconvolution software (Note that these acquisition settings were used for all experiments depicted in the manuscript, unless otherwise stated). Quantification of centriole number and length was performed in at least 50 mitotic cells, for each of the cell lines tested, using our in-house developed algorithm (see description below).

**Secondary immunofluorescence screening.** Cells were stained with DAPI, centrin-1 (1/1000 dilution, clone 20H5, Millipore), CP110 (1/250 dilution, homemade) and α-tubulin (1/100 dilution, YL1/2 MCA776, Serotec) antibodies (for more details, Immunofluorescence staining section). Confocal image stacks were acquired on a Yokogawa CSU-X1 Spinning Disk confocal scan head, coupled to a Nikon Ti microscope, using a Nikon ×100 Apo TIRF 1.49 NA Oil immersion objective with the ×1.5 auxiliary magnification. 405, 491, 561 and 642 nm wavelength lasers were used to excite DAPI, Alexa 488, Rhodamine and Alexa 647, respectively. Images were acquired with a Photometrics Evolve 512 EMCCD camera through Metamorph acquisition software (Molecular Devices). Images were taken as Z-stacks in a range of 10–14 µm, with a distance between planes of 0.2 µm.

Deconvolution in three spatial dimensions was performed with the Parallel Iterative Deconvolution plugin of ImageJ, using the Wiener Filter Preconditioned Landweber algorithm with the following parameters: weiner = 0.001 (expressed as a fraction of the largest Fourier coefficent of the PSF), low = 1, maximum number of iterations = 25, terminate = 0.01 (terminate if mean delta less than this value). This methodology requires a normalised point spread function (PSF), which was previously created with the Diffraction PSF 3D PlugIn of ImageJ with the parameters: Index of refraction of the mounting median = 1.46 (ProLong® Gold Antifade Reagent, Molecular Probes®); numerical aperture NA = 1.44, wavelength = 461, 519, 590, 667 nm for DAPI, AlexaFluor-488, Rodamine Red-X, and AlexaFluor-647, respectively; image pixel spacing = 86.7 nm, slice spacing ($z$) = 200 nm, width = 64 pixels, height = 64 pixels.

Quantification of centriole number and length was performed in at least 50 mitotic cells, for each of the cell lines tested, using our in-house developed algorithm (see description below).

**Semi-automated quantification of centriole number and length.** Centrioles located at the vicinity of mitotic spindles were identified with the following algorithm encoded in C/C++ using OpenCV 2.4.3 libraries (source code available on request to A.G.). The segmentation of the mitotic figures from interphasic cells was performed using the following procedure: calculate the maximum intensity projections of the DAPI and alpha-tubulin (rhodamin) signals; MaxB and MaxR, respectively (step 1.1). Smooth MaxB and MaxR with a Gaussian kernel, $\sigma = 1.15$ µm (step 1.2). Calculate the per-element product of MaxB and MaxR, which we named as CM, the correlation matrix (step 1.3). This CM serves to determine the regions containing mitotic figures, characterised by having both a bright DAPI (due to DNA condensation) and a bright alpha-tubulin (spindle) staining. Segmentate the brightest objects of CM by the Otsu's method[68] (step 1.4). Calculate the signal to noise ratio ($S/N$) of CN, where $S$ corresponds to the mean pixel intensity of the object class and $N$ corresponds to the mean value of the pixels of non-object class (step 1.5). Keep the objects with a $S/N$ higher than 2.5. Otherwise segmentate the objects from the CM by an intensity criterion: Calculate the mean (CM$_{mean}$) and standard deviation (CM$_{sd}$) of CM and classify each pixel of CM to be part of the object class (if its value is higher than $C_{mean} + 6 \times C_{sd}$), or to be part of the background class (if its value is less or equal or less than $C_{mean} + 6 \times C_{sd}$) (step 1.6). Dilate the remaining objects with a circular kernel, with $r = 2.2$ µm (step 1.7). Apply a circularity criterion (Circ) over the objects to discharge edgy objects. The Circ coefficient, which we define as Circ = $(H_{area} - O_{area})/H_{area}$, is a non-dimensional quantity that relates the area of a given object ($O_{area}$) to the area of its convex hull ($H_{area}$), calculated with the Sklansky's algorithm (Sklansky, 1982), with $H_{area} \leq O_{area}$. An edgy object will have Circ close to 1 and a circular object will have a Circ close to 0. Keep object with a Circ < 0.2 (step 1.8). Eliminate small objects (delete if $O_{area} < 400$ µm$^2$), (step 1.9). Dilate the remaining objects with a circular kernel, with $r = 2.2$ µm, to include regions closer to the spindle (step 1.10). Each remaining 2D contour object will contain a mitotic figure and any centriole located within a vicinity of 2.2 µm (step 1.11).

The segmentation of centrioles located at the vicinity of mitotic figures was performed using the following procedure: Calculate the maximum intensity projection, MAX$_{centrin}$ and MAX$_{CP110}$, of each centrin and CP110 channel, respectively (step 2.1). Note that, for the primary screening, we only used a single centriolar marker, namely centrin. As a consequence, for the primary screening, we used only MAX$_{centrin}$, while, for the secondary screening, both MAX$_{centrin}$ and MAX$_{CP110}$ were used. Segmentate, from MAX$_{centrin}$, and MAX$_{CP110}$, the regions

surrounded by the contours found in step 1.11, which we call $\text{ROICell}_{centrin}$ and $\text{ROICell}_{CP110}$ (step 2.2). For each $\text{ROICell}_{centrin}$ and $\text{ROICell}_{CP110}$ matrices, perform a bicubic interpolation overall ROICell pixels, with a size factor of 8, to create the ROI.8xyCell matrix (step 2.3). This interpolation step helps to better define the contours of the centrioles, which will be segmentated in the following steps. Segmentate the brightest objects of each ROI.8xyCell by the Otsu method. Use the Mander's overlap coefficient to identify colocalising objects between $\text{ROI.8xyCell}_{centrin}$ and $\text{ROI.8xyCell}_{CP110}$ matrices (the co-localisation step was omitted for the primary screening) (step 2.4). Each of the remaining objects, which we stored on a given $\text{ROI.8xCent}_i$ matrix, will contain the maximum intensity projections of the centrin channel, each corresponding to the 2D projection of a putative centrosome. Delete any $\text{ROI.8xCent}_i$ object smaller than $0.1 \ \mu m^2$, and bigger than $1 \ \mu m^2$ (step 2.5). Identify the corresponding $\text{ROI.8xCent}_i$ pixels in all z-stacks of ROI.8xyCell (step 2.6). Interpolate the corresponding z-planes with a bicubic interpolation, and segmentate the pixels associated to the putative $\text{centrosome}_i$ in 3D from the resulting matrix using the same threshold value calculated in step 2.4. Store the segmentated objects on the $\text{ROI.8x.3DCent}_i$ matrix. Perform a principal component analysis over any object of $\text{ROI.8x.3DCent}_i$ to calculate its eigenvectors (step 2.7). Create a maximum intensity projection over the first eigenvector of each $\text{ROI.8x.3DCent}_i$ object and calculate its inflexion points (with a Sobel operator, step 2.8). For each object of $\text{ROI.8x.3DCent}_i$ create a maximum intensity projection over the plane created by its first and second eigenvectors (step 2.9). For each $\text{ROI.8x.3DCent}_i$ object, use a perpendicular plane, shaped by its second and third eigenvectors, to split it at the inflexion points found in step 2.8 (step 2.10). Repeat iteratively steps 2.7 to 2.9 over each resulting object until any single object of $\text{ROI.8x.3DCent}_i$ becomes totally segmentated (step 2.11). The resulting objects obtained from steps 2.9 to 2.11 will be considered as single centrioles (step 2.12). The 3D length of each resulting centriole is finally calculated from its first eigenvalue.

In summary, steps 1.1 to 1.11 allow to identify mitotic figures in a background of interphasic cells; steps 2.1 to 2.6 allow to identify the centrosomes located in the vicinity of a given mitotic spindle; and steps 2.7 to 2.11 allow to isolate, count and measure the length of single centrioles. Our current implementation of these algorithms allows to perform manual curation at steps 1.4 and 2.10 (Fig. 1b). The overall analysis of centriole number and length is depicted in Figs 2 and 3, respectively. Two different population parameters, namely, the mean and the variance-to-mean ratio (VMR), were used to compare centriole length between the different cell lines Fig. 3b.

**Code availability**. All the codes of the algorithm are available upon request to A.G. (adanog@ibt.unam.mx).

**Transmission electron microscopy**. Pellets of cells were fixed using 2.5% glutaraldehyde, in a 1× PBS solution, for 30 min at room temperature and postfixed with 1% Osmium tetroxide ($OsO_4$) solution for 1 h at 4 °C. The samples were stained en bloc with 2% uranyl acetate for 20 min and dehydrated in a graded series of ethanol. The samples were treated with propylene oxide for 15 min, followed by a 1:1 mix of propylene oxide and resin for 15 min and a pure resin incubation for 1 h. The samples were polymerised at 60 °C for 24 h. Serial sections (60–120 nm) were taken on a Leica Reichert Ultracut S ultramicrotome, collected on formvar-coated copper slot grids, and stained with uranyl acetate and lead citrate. The sections were analysed at 100 kV on a Hitachi 7650 TEM. Centriole length quantification was performed from TEM images of longitudinal sections of centrioles using Image J software.

**CPAP overexpression in T47D, SF268 and U2OS cells**. U2OS, T47D and SF268 cells were transfected with either peGFP-C hCPAP (kind gift from Juliette Azimzadeh), or pLJ499-EYFP plasmid (kind gift from Lars Jansen's laboratory), using the Neon® Transfection system. The following parameters were respectively used for T47D, SF268 and U2OS: 1700, 1300 and 1200 V Pulse voltage, 20, 30 and 10 ms pulse width and 1, 1 and 4 pulses were used.

For U2OS, cells were either pelleted for further TEM processing (see Transmission electron microscopy section above) or incubated for 1 h on ice to induce the depolymerisation of cytoplasmic microtubules after 48, 72 or 96 h of overexpression. Subsequently, cells were stained with DAPI, CEP135 (1/500 dilution, kind gift from Ryoko Kuriyama) and acetylated tubulin (1/500 dilution, Sigma, Ref. T7451) antibodies (Immunofluorescence staining section for more experimental details).

The number and length of centrioles was quantified using the maximum intensity projection of the acetylated tubulin staining. Only centrioles positive for at least two of the centriolar markers used (CPAP, acetylated tubulin or CEP135) were analysed. Three independent experiments were performed. For the centriole length quantification, 100 centrioles were measured per condition and per experiment. A Mann–Whitney test was used to compare centriole length between different conditions (see Supplementary Fig. 9c); a two-tailed 5% significance level was considered as statistically significant ($p < 0.05$). To obtain the percentage of cells with >4 centrioles, 100 cells were counted per condition and per experiment. Similar experiments were performed for the centrinone B assay (see Supplementary Fig. 9d), except that cells were incubated with centrinone B (500 nM) or DMSO

(negative control) 24 h prior transfection and throughout the entire time course of the experiment.

For T47D and SF268 cells, cells were either incubated for 2 h on ice prior to fixation to induce the depolymerisation of cytoplasmic microtubules (PCM quantification) or directly fixed with cold methanol (10 min at −20 °C, centriole number and chromosome segregation defects quantifications) after 48 or 96 h overexpression, respectively. For PCM quantification, we decided to restrict our analysis to one PCM component (pericentrin), as both pericentrin and γ-tubulin quantifications gave similar results in MDA-MB-435, HOP-62 and RPE cells (Fig. 6 and Supplementary Fig. 11). Therefore, T47D and SF268 cells were stained with DAPI, acetylated tubulin (1/500 dilution, Sigma, Ref. T7451) and pericentrin (1/250 dilution, ref. AB44489 Abcam) antibodies. For centriole number and chromosomal instability quantifications, cells were stained with DAPI, alpha-tubulin (1/200, ref. AB18251, Abcam) and centrin-1 (1/1000 dilution, clone 20H5, Millipore) antibodies.

For PCM quantification, we restricted our analysis to 48 h samples as centriole structures really become very defective upon CPAP overexpression making the analysis too difficult at later time points. Even with this restriction, centriole structures were too defective to apply the macro designed for PCM quantification in MDA-MB-435, HOP-62 and RPE (see below). As an alternative, we measured pericentrin fluorescence intensities at each centrosome using Image J software. Briefly, we created the sum intensity z-projections and measured the raw integrated density of regions of interest (ROI) around each centrosome. To correct for background fluorescence, we measured the raw integrated densities of region of same size of the ROI that were outside centrosomes but within the cell. Three independent experiments were performed with around 30 centrosomes accounted per condition and per experiment. Note that the quantification was performed in interphasic cells as the mitotic content was too low. A Mann–Whitney statistical test was used to compare pericentrin densities between different conditions (Fig. 6b).

For both centriole number and chromosomal instability quantifications, we focused our analysis on the 96 h sample to study the long-term effect of centriole length deregulation. For centriole number quantification (Fig. 5b), centrin was used as a readout to count centrioles in mitotic cells. Three independent experiments were performed (between 80 and 140 cells were counted per condition and per experiment). A one-tailed unpaired $t$ test with Welch's correction was used to compare centriole number between different conditions (see Fig. 5b).

To assess chromosomal instability, we first quantified the percentage of multipolar mitosis during prometaphase and metaphase using α-tubulin staining (Fig. 6e). Three independent experiments were performed (between 30 and 87 mitosis were accounted per condition and per experiment). A one-tailed z score test was performed the compare the proportion of multipolar mitosis between conditions. Second, we scored different types of chromosome segregation defects during anaphase-telophase: polar asymmetry (chromosome(s) in the vicinity of one of the two spindle poles at anaphase onset), DNA/chromosome bridge ("string-like" connection between the two masses of segregating chromosomes), lagging chromosome (chromosomes that lag between the two masses of segregating chromosomes), DNA/chromosome protrusion (improperly aligned chromosome/ DNA that probably results from bridges) and multipolar mitosis (Fig. 6f and Supplementary Fig. 12c). Three independent experiments were performed (between 38 and 55 mitosis were accounted per condition and per experiment). A one-tailed zscore test was performed the compare the proportion of chromosome segregation defects between conditions.

**Structured illuminated microscopy**. MDA-MB-435, HOP-62 and RPE-1 cells were incubated 2 h on ice to induce the depolymerisation of cytoplasmic micro-tubules, and were subsequently stained with DAPI, acetylated tubulin (1/1000 dilution, Ref. T7451, Sigma) and STIL (1/500 dilution, Ref. A302-441A, Bethyl laboratories) antibodies (for more details, Immunofluorescence staining section). Super-Resolution images were acquired using an LSM780 ELYRA PS1 (SR-SIM) microscope (Zeiss), and a ×63 objective. Images were collected with five grid rotations, using a 1.513 refractive index oil at room temperature. Post-acquisition processing of images, including structured illumination reconstruction, was done using Zen (Zeiss).

**Age-related centriole staining**. MDA-MB-435 cells were incubated for 2 h on ice to induce the depolymerisation of cytoplasmic microtubules. Subsequently, cell lines were stained with DAPI, acetylated tubulin (1/1000 dilution, Ref. T7451, Sigma), STIL (1/500 dilution, Ref. A302-441A, Bethyl laboratories) and eventually CEP164 (1/700 dilution, Ref. SC240226, Santa Cruz Biotechnology) antibodies (for more details, Immunofluorescence staining section). The number of overly long centriole(s) per cell (Supplementary Fig. 5b) was determined using acetylated tubulin staining, as a readout for centriole length, and the age of centrioles was assessed using STIL staining (absence = mother, presence = daughter, Supplementary Fig. 5c). In addition, the presence of ectopic procentriole was accounted when overly long centrioles possessed at least two STIL dots along their length (Supplementary Fig. 10c). Centrioles were considered as overly long when they were at least approximately twice the size of normal-length centrioles. Three independent experiments were performed with at least 68 cells displaying long centriole(s) analysed per experiment.

**Cilia staining**. MDA-MB-435 and HOP-62 cells were incubated for 2 h on ice to induce the depolymerisation of cytoplasmic microtubules. Subsequently, cell lines were subjected to an IF assay using DAPI and two different marker combinations: (i) centrin-2 (1/500 dilution, Ref. sc-27793-R, Santa Cruz Biotechnology) and acetylated tubulin antibodies (1/500 dilution, Sigma, Ref. T7451), to visualise the long centrin-positive rod-like structures and stable MTs present in centriole and cilia, respectively; (ii) centrin-1 (1/1000 dilution, clone 20H5, Millipore) and ARL13B (1/500 dilution, kind gift from T. Caspary), to label the long centrin-positive rod-like structures and cilia, respectively (for more experimental details, Immunofluorescence staining section). The visual examination of the presence of acetylated tubulin, or ARL13B, on the long centrin-positive rod-like structures was performed on an Applied Precision DeltavisionCORE system (for representative pictures see Supplementary Fig. 4). These structures were considered as overly long when they were at least approximately twice the size of normal-length centrioles. Three independent experiments were performed, with at least 30 overly long centrioles analysed per condition and per experiment.

**PCM components quantification**. RPE-1, MDA-MB-435 and HOP-62 cell lines were incubated 2 h on ice to induce the depolymerisation of cytoplasmic microtubules, and were subsequently stained with DAPI and two different marker combinations to visualise both centrioles and PCM: (i) centrin-2 (1/500 dilution, Ref. sc-27793-R, Santa Cruz Biotechnology) and γ-tubulin (1/25 dilution, ref. GTU88, Sigma), (ii) centrin-1 (1/1000 dilution, clone 20H5, Millipore) and pericentrin (1/250 dilution, ref. AB44489 Abcam). For more experimental details, please see the Immunofluorescence staining section.

For the PCM quantification, acquired data (see "Primary immunofluorescence screening" section for technical details) were processed and quantified with Fiji (ImageJ distribution[69]). Centrosome detection was performed on a Maximum Intensity projection, using Yen automatic threshold algorithm, to get a minimum threshold value to be used as a reference for 3D Object Counter plugin[70].

Total centrin and pericentrin/γ-tubulin sum intensities of all voxels of a centrosome were obtained by 3D Object Counter's Integrated Density parameter. A local background value was calculated for each centrosome by creating the average intensity projection and multiplying this value by the number of voxel of each 3D object to get the contribution of the background/noise to the total measured Integrated Density. The Centrosome fluorescence signal was then calculated by subtracting the background integrated density to the object total integrated density. The centrin and pericentrin or γ-tubulin integrated densities were plotted against each other to investigate if centriole length and PCM content are correlated in the two cell lines displaying centriole over-elongation, MDA-MB-435 and HOP-62 (Supplementary Fig. 11b, c). Three independent experiments were performed, and at least 250 centrosomes were quantified per cell line and per PCM component. The Spearman correlation coefficient was calculated and the two parameters were considered correlated if $p < 0.05$.

To further push the analysis, we took advantage of the fact that mitoses were often visually displaying asymmetric centriole content in MDA-MB-435 cell line to directly compare, at the single cell level, the PCM recruitment ability of centrosomes containing overly long centrioles versus centrosomes with normal-length centrioles. Please note that we used RPE-1 and MDA-MB-435 cells with only normal-length centrioles as controls. For mitosis with normal-length centrioles (named "Normal-length"), the pericentrin or γ-tubulin integrated density of the pole with higher centrin intensity was normalised to the integrated density of the pole with lower centrin intensity. For mitotic cells with an asymmetric content of centrioles, the pericentrin or γ-tubulin integrated density of the pole containing longer centriole(s) was normalised to the integrated density of the other pole. Centrioles were considered as overly long when they were at least twice the size of normal-length centrioles. Three independent experiments were performed (between 37 and 62 centrosomes were quantified per condition and per experiment). One-tailed unpaired t tests with Welch's correction were used to compare samples.

**MT regrowth assays**. RPE-1 and MDA-MB-435 cells were incubated 2 h on ice to induce the depolymerisation of cytoplasmic microtubules. After these 2 h, freshly-warmed media was added to the cells and the plates were put at 37 °C for 45 s to allow MTs regrowth and subsequently fixed and stained with DAPI, centrin-1 (1/1000 dilution, clone 20H5, Millipore) and alpha-tubulin (1/200, ref. AB18251, Abcam) (for more experimental details, Immunofluorescence staining section). The α-tubulin and centrin fluorescence intensities were measured at each centrosome using Image J software. Briefly, mitotic cells were manually selected and the area outside the selection was erased using the "clear outside" function. To correct for background fluorescence, the min intensity z-projections were created allowing for average intensity value measurement. These values were subtracted from each stack and, then, the sum intensity z-projections were created and the raw integrated densities of region of interest around each centrosome were measured.

To directly investigate the effect of the variable "centriole length" on the ability to recruit PCM, the α-tubulin raw integrated density of the pole with higher centrin intensity was normalised to the raw integrated density of the pole with lower centrin intensity in mitosis with normal-length centrioles (named "Normal-

length"). For mitotic cells with an asymmetric content of centrioles, the α-tubulin integrated density of the pole containing longer centriole(s) was normalised to the integrated density of the other pole. Note that centrioles were only considered as overly long when they were approximately at least twice the size of normal-length ones. Three independent experiments were performed (between 30 and 82 centrosomes were accounted per condition and per experiment). A one-tailed unpaired t test with Welch's correction was used to compare the different conditions.

**Quantification of chromosome segregation defects**. MDA-MB-435 cells lines stained with DAPI, to visualise DNA, and CENPB antibodies (1/200 dilution[71]), to label centromeres. For centrioles, a double staining was performed using centrin-1 (1/1000 dilution, clone 20H5, Millipore) and CP110 (1/250 dilution, homemade) antibodies. The green channel was used to visualise both centromeres and centrioles (centrin antibody), since these structures have distinct locations in the cell (for more experimental details, Immunofluorescence staining section). Images of anaphases and telophase were taken as Z-stacks on an Applied Precision DeltavisionCORE system using a ×100 1.4 NA Oil immersion objective and were blindly deconvolved with Applied Precision's SoftWorx software. Subsequently, these images were scored for different types of chromosome segregation defects: polar asymmetry (chromosome(s) in the vicinity of one of the two spindle poles at anaphase onset), DNA/chromosome bridge ("string-like" connection between the two masses of segregating chromosomes), lagging chromosome (chromosomes that lag between the two masses of segregating chromosomes) and DNA/chromosome protrusion (improperly aligned chromosome/DNA that probably results from bridges).

Three independent experiments were performed, in which we analysed only cells displaying normal-length centrioles or cells exhibiting overly long centriole(s) at one spindle pole (200 cells were counted per experiment). Note that centrioles were only considered as overly long when they were approximately at least twice longer than normal-length ones. A z score test was used to compare the overall proportions of anaphases and telophases with chromosome segregation defects between the two groups of cells, $p = 0.0001$.

**Data availability**. The data that support the findings of this study are available within the Article and Supplementary Files, or available from the authors upon request.

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

## Acknowledgements

We thank the Cancer and Centrosome Consortium of the Harvard Medical School Portugal Program, the Harvard Medical School Portugal Program members and directors, and all members of M.B-D. lab for fruitful discussions. We are thankful to Jadranka Loncarek and Joana Vaz for help with experiments and critical reading of the manuscript. We are also grateful to Rita Fior, Sérgio Dias and Filipe Leal for critical reading of the manuscript. We are deeply grateful to all the people that provided us with the cell lines. G.M. and A.G. were funded by the FCT-Harvard Medical School Program Portugal grant (HMSP-CT/SAU-ICT/0075/2009) and individual FCT post-doctoral fellowships (SFRH/BPD/98439/2013 and SFRH/BPD/82420/2011, respectively). The M.B-D. laboratory is supported by IGC, an EMBO installation grant, ERC grant ERC-2010-

StG-261344, FCT grants (FCT Investigator to M.B-D., POCI-01-0145-FEDER-016390 and PTDC/BIM-ONC/6858/2014) and a FCT-Harvard Medical School Program Portugal grant (HMSP-CT/SAU-ICT/0075/2009).

## Author contributions

G.M. and M.B-D. designed and analysed all experiments and wrote the manuscript. S.A.G., B.V., G.M. and I.F. prepared samples for both screenings. A.G. designed the algorithm to quantify centrioles and curated all galleries with G.M. G.M., S.M., P.M. and E.M.T. did the TEM experiments. M.M. optimised the protocol for centriole detection in human tissue samples. J.P. and A.F.V. performed, analysed and provided expertise on the human breast carcinoma results. G.M. performed most of the cell biology experiments. K.D. helped with the characterisation of long centrioles. M.E.F. and S.C.J. acquired and processed the super-resolution pictures. D.P. and S.A.G. started the screen and provided expertise on centrosome and cancer. N.P.M. designed the workflow and macro for PCM quantifications. B.P.d.A. and N.L.B-M. analysed and provided expertise on the screening results. All authors read and discussed the manuscript.

## Additional information

**Competing interests:** The authors declare no competing interests.

