## [Peer Review File(PDF 546 kb) · Nature Communications]

Reviewers' comments:

Reviewer #1 (Remarks to the Author):

In this manuscript, the authors used a semi-automated screening method to measure centriole abnormalities including centriole number and length in the NCI-60 panel of cancer cell lines and patient samples. Their results showed that centriole amplification and length deregulation are common features in a number of cancer cell lines, and aggressive breast and colon cancer exhibit high levels of centriole amplification. They also found that centriole over-elongation could drive centriole amplification via centriole fragmentation and ectopic procentriole formation in certain cancer cells. Further studies revealed that these extra-long centrioles could lead to produce over-active centrosomes that nucleate more microtubules and recruit more PCM proteins. Together, they conclude that deregulation of centriole length in cancer cells promote centriole amplification and abnormal chromosome segregation. In general, the manuscript is interesting and contains an impressive amount of data using a wide variety of techniques. However, this work lacks direct evidence to support the concept that deregulation of centriole length promote centriole amplification and chromosome missegregation. Although the work lack direct evidence and still leaves many open questions, the study is suitable for publication in nature communications after dealing with the following important questions:

Major points:

- 1). The authors showed that overly long centrioles recruit more PCM proteins (gamma-tubulin and pericentrin; Fig. 6A,B), nucleate more microtubules (MT) (Fig.6C, D), and enhance abnormal chromosome segregation (Fig. 6E, F) in cancer cells. These findings lead the authors to propose that extra long centrioles form over-active centrosomes and promote abnormal chromosome segregation. However, the data they provided are indirect and restricted mainly to a MDA-MB-435 cancer cell line. The authors should also examine the MT nucleation activity and chromosome missegregation using two or more cancer cell lines. Importantly, the authors need to provide direct evidence to demonstrate that centriole over-elongation could trigger PCM recruitment and chromosome missegregation in CPAP transfected cells (the authors may use cancer cell lines with no or low centriole over-elongation for CPAP transfection study), in which the overly long centrioles could be induced by overexpressing CPAP.
- 2). The statement that centriole length deregulation drives centriole amplification in cancer cells is overemphasized, since only one cancer cell line, MDA-MB-435, was used in this study (Fig. 5B). The authors should include data using two or more cancer cell lines. Furthermore, the authors need to provide statistics for Fig. 5B.
- 3). The authors need to provide statistics for the total intensity of gamma-tubulin and pericentrin associated with overlong centrioles in MDA-MB-435 and HOP-62 cells (Fig. 6A).

Minor points:

- 1). The enlarged photos in Fig. 1B should provide a scale bar.
- 2). Fig. 6C and Fig. S6C need to provide a scale bar.

Reviewer #2 (Remarks to the Author):

In this study Marteil and colleagues perform an extensive study of 60 cancer cell lines in terms of centrosome number and centriole size. This study is based on an automated image-analysis setup that allowed the authors to characterize centriole numbers and size in mitotic cells. The authors find that abnormal centrosome numbers and an elongated centriole size is a frequent characteristic of cancer cells. Based on their results they further propose that over-elongated centrioles can lead to centriole fragmentation, ectopic pro-centriole formation, and increase microtubule nucleation

capacity, and that these defects lead to an elevated rate of chromosome segregation errors.

On the plus side this study addresses an important biological question, which is the extent of centrosome abnormalities in cancer tissues. While centrosomes are generally thought to be abnormal in cancer cells, such a systematic study provides a much more definite answer. Moreover the author describe an novel type of centrosome defect, centriole over-elongation, and link it to the formation of multipolar spindles and chromosome segregation errors, which makes this study novel, original and of strong interest to the mitosis/centrosome field. On the negative side, this study suffers from a number of major weaknesses (see below), which partially undermine the major conclusions of this study. Therefore, even though this manuscript has a strong potential, it is too preliminary for publication, and it would benefit from a major set of revisions.

Major points:

1.) Screening for centriole amplification: To characterize centriole amplification, the authors analyzed 50-60 cancer cells from breast, brain, colon, blood, lung, prostate, kidney, ovaries and skin and compared it to 1 control non-cancer cell line, the retina epithelial cells RPE-1. If at least 16% of the cells displayed more than 4 centrioles, cells were classified as having amplified centriole numbers. There are two fundamental problems with this screen. First, the sample size is very low. With a minimum of 50 cells and a cut-off probability of 16% one obtains a 95% confidence interval of $\pm 10\%$, which means that the classification of many cell lines is uncertain. The sample size should be increased. Second, we do not know what is the variability of centriole numbers in non-cancer cells, since the authors only used 1 cell line (RPE-1) as a negative control. Moreover, this cell line originates from a tissue (retina), which is different from all the tested cancer cell line tissues. Ideally, the authors should test one non-cancerous cell line from every tissue studied in this paper, as this would provide a comparison within the same tissue, and would define the variation in centriole numbers one should expect. Improving these two points is essential, since the authors use this classification throughout the study.

2.) Link between centriole length and centriole amplification: the authors postulate the elongated centrioles might contribute to centriole amplification. This is supported by EM-data, which show that elongated centrioles can fragment and/or lead to ectopic pro-centriole formation. However, the crucial correlation between the two phenotypes is Figure 5A, which suggests that cells with elongated centrioles are more likely to also have an elevated number of centrioles. However, if one takes the number given in Figure legend 5A for every category and performs a Chi-square test on a two-by-two table, one finds that correlation between elevated centriole length and centriole numbers is not statistically significant (p-value of 0.069 for a one-sided Chi-square test). Although there is a tendency, the authors can on the basis of their numbers not conclude that an elongated centriole length leads to centriole amplification.

3.) Microtubule nucleation capacity: in Figure 6C and D the authors test whether the elongated centriole length leads to a higher microtubule nucleation capacity. Although the quantification in Figure 6D strongly supports this hypothesis, the picture shown in figure 6C is of poor quality, and one cannot recognize the typical microtubule aster that should emanate from the centrosomes in a microtubule nucleation assay. The authors should therefore improve this assay to obtain much more convincing images reflecting microtubule nucleation.

4.) Centriole elongation and chromosome segregation: based on Figure 6E and F the authors propose that the elongated centrioles contribute to chromosome segregation errors. The quantification in Figure 6F is convincing, but one worry is that the two phenotype are only linked by a correlation. One could imagine that the MDA-MB-435 cell line is composed to two cell populations, one with elongated centrioles, the other without, and that the first population has a higher chromosome segregation error rate. The situation would be very different if the authors could suppress the segregation errors by reducing microtubule nucleation in these cells (gamma-tubulin or pericentrin depletion?). Moreover, it would be helpful to the reader if the authors would quantify the type of errors seen in both cell populations, since changes in microtubule dynamics

are usually associated with lagging, merotelic chromosomes, and not so much with the chromosome bridges depicted in the immunofluorescence images of Figure 6E. Finally, it would be helpful if the authors would speculate on possible mechanisms by which changes in microtubule nucleation might lead to chromosome segregation errors.

Reviewer #3 (Remarks to the Author):

The manuscript *Deregulation of Centriole Length is Widespread in Cancer and Promotes Centriole Amplification and Chromosome Missegregation*

Abnormal numbers of centrioles and centrosomes and altered centrosome functions contribute to high rates of chromosome missegregation and are frequent in cancer. The authors of this manuscript decided to closely characterize this phenomenon by detailed analysis of centriole numbers and shape in the cancer cell line collection NCI-60. Their systematic survey of cancer cell lines revealed that many cells have high number of centrioles and many centrioles are abnormally elongated. These abnormalities correlated with the centrosome microtubule nucleation activity and with frequency of chromosome segregation error. The work is carefully executed and the manuscript is well written, but lacks novelty and original insight. Most of the thoughts, ideas and findings from this manuscript have been previously shown, although not in 60 different cell lines. Therefore, this article seems appropriate for a more specialized journal, such as *Cancer Research* or *Oncotarget*.

Several remarks:

1. Although the authors look at multiple breast tumor samples, the main body of work has been done in culture cell lines. The question to what degree is this informative about the real situation in cancer is not even discussed.
2. It is interesting that even the most affected cell lines show centriole amplification only in a minority of the cells. Why is that? How these cells differ? What causes the heterogeneity? This is not discussed at all either.
3. Figures 6E – what is stained with the red signal on the right panel? Is it CP110 as in the left panel? Why is it so different from the left panel?
4. The relationship between centriole changes and chromosome segregation errors is rather vague. It would be more useful to look whether there is indeed a strong correlation of the centriole changes and CIN levels. The number of chromosomes in karyotype and its range is known for the NCI60 cells. More careful analysis correlating the centriole numbers, chromosome numbers and the range and variance in the chromosome number might be more informative.
5. The same is true for “ploidy deregulation”. This is a very vague term, as is the entire paragraph about ploidy and centriole numbers.
6. How can the altered centriole numbers cause anaphase bridges?

Point by point answers to referees

Reviewer #1

In this manuscript, the authors used a semi-automated screening method to measure centriole abnormalities including centriole number and length in the NCI-60 panel of cancer cell lines and patient samples. Their results showed that centriole amplification and length deregulation are common features in a number of cancer cell lines, and aggressive breast and colon cancer exhibit high levels of centriole amplification. They also found that centriole over-elongation could drive centriole amplification via centriole fragmentation and ectopic procentriole formation in certain cancer cells. Further studies revealed that these extra-long centrioles could lead to produce over-active centrosomes that nucleate more microtubules and recruit more PCM proteins. Together, they conclude that deregulation of centriole length in cancer cells promote centriole amplification and abnormal chromosome segregation. In general, **the manuscript is interesting and contains an impressive amount of data using a wide variety of techniques.** However, this work **lacks direct evidence to support the concept that deregulation of centriole length promote centriole amplification and chromosome missegregation. Although the work lack direct evidence and still leaves many open questions, the study is suitable for publication in nature communications after dealing with the following important questions:**

We were happy that Reviewer 1 liked our manuscript and recommended it for publication in *Nature Communications*. Below we describe our attempts to address his/her concerns regarding the lack of direct evidence supporting the conclusion that centriole over-elongation, in part by enhancing PCM recruitment, promotes chromosome segregation errors.

Major points:

1). The authors showed that overly long centrioles recruit more PCM proteins (gamma-tubulin and pericentrin; Fig. 6A,B), nucleate more microtubules (MT) (Fig.6C, D), and enhance abnormal chromosome segregation (Fig. 6E, F) in cancer cells. These findings lead the authors to propose that extra long centrioles form over-active centrosomes and promote abnormal chromosome segregation. However, the data they provided are indirect and restricted mainly to a MDA-MB-435 cancer cell line. **The authors should also examine the MT nucleation activity and chromosome missegregation using two or more cancer cell lines.**

Importantly, the authors need to provide **direct evidence to demonstrate that centriole over-elongation could trigger PCM recruitment and chromosome missegregation in CPAP transfected cells (the authors may use cancer cell lines with no or low centriole over-elongation for CPAP transfection study),** in which the overly long centrioles could be induced by overexpressing CPAP.

To address the first part of reviewer 1 comments, we selected two additional NCI-60 cell lines with centriole length deregulation, HOP-62 and H23. Unfortunately, the number of mitotic cells with a clear and visible asymmetry in centriole content (i.e. on mitotic pole with long centriole(s) and the other pole with normal-length ones) was not sufficient to

perform trustable quantifications of MT nucleation activity and chromosome missegregation.

We also performed the complementary experiment suggested by the reviewer (CPAP overexpression) to evaluate centriole elongation, PCM recruitment and chromosome missegregation. These experiments were informative. We transiently overexpressed CPAP in two cancer cell lines from the NCI-60 panel that exhibit low levels of centriole amplification and over-elongation, T47D and SF268. First, we confirmed that elongated centrioles contain more PCM than normal-length centrioles, therefore confirming that the presence of elongated centrioles leads to the formation of larger centrosomes (Figure 6B). We then observed that the proportion of prometaphases-metaphases displaying multipolar spindles - a known source of chromosome instability (Ganem et al, 2009; Silkworth et al, 2009) - statistically increases upon CPAP overexpression in both cancer cell lines (Figure 6E). Finally, we saw that the proportion of anaphase-telophase displaying chromosome segregation defects (polar asymmetry, DNA/chromosome bridge, lagging chromosomes, misaligned chromosomes and multipolar cells) is statistically enhanced upon CPAP overexpression in both cell lines (Figures 6F and S11C). Altogether these results show that induced centriole elongation triggers enhanced PCM recruitment and chromosome segregation defects in cancer cell lines. We thank the reviewer for this suggestion.

2). The statement that centriole length deregulation drives centriole amplification in cancer cells is overemphasized, since only one cancer cell line, MDA-MB-435, was used in this study (Fig. 5B). **The authors should include data using two or more cancer cell lines. Furthermore, the authors need to provide statistics for Fig. 5B**

To further test our hypothesis, we determined if centriole fragmentation and ectopic procentriole formation occurred in an additional cancer cell line displaying elongated centrioles, HOP-62. While we did not observe ectopic procentriole formation in this cell line, we could clearly observe, by Structured Illumination Microscopy, that overly long centrioles can fragment, supporting that centriole fragmentation drives centriole amplification (please see Figure S9A (iii)). Another experiment already present in the original version of our manuscript further supports the presence of centriole fragmentation (new Figure S8). Briefly, while inducing centriole over-elongation (CPAP over-expression in U2OS cells), we prevented centriole biogenesis by inhibiting the master regulator of this process, PLK4, using the specific chemical inhibitor centrinone B (Wong et al, 2015). We observed that cells treated with centrinone B and over-expressing CPAP still display centriole amplification, unlike cells solely treated with centrinone B (Chi-square test, $p=0.003$, Figure S8D). These results demonstrate that centrosome amplification upon CPAP overexpression is partially independent of centriole biogenesis, therefore confirming that overly long centrioles fragment.

Regarding the statistics for old Figure 5B (new Figure 5C), we observed that, on average, 4% of overly-long centrioles can nucleate extra-procentrioles in MDA-MB-435 cell line (Figure S9B), therefore confirming that ectopic procentriole formation can partially contribute to centriole amplification. For centriole fragmentation, we can only observe this phenotype by SIM (after reconstruction) and by transmitted electron microscopy (TEM). This limitation, together with the fact only a subpopulation of centrioles is over-elongated, prevented us from doing quantifications with a statistical significant sample number.

To further substantiate the direct link between centriole elongation and amplification, we transiently overexpressed CPAP in two cancer cell lines from the NCI-60 panel that exhibit low levels of centriole amplification and over-elongation, T47D and SF268. As shown in new Figure 5B, the percentage of mitotic cells with more than 4 centrioles significantly increases upon induced centriole elongation. Altogether, our new results clearly reinforce

our statement that centriole length deregulation promotes centriole amplification in cancer cells (please note that we have modified our title and abstract accordingly).

3). The authors need to provide statistics for the total intensity of gamma-tubulin and pericentrin associated with overlong centrioles in MDA-MB-435 and HOP-62 cells (Fig. 6A).

We agree that this quantification would greatly improve the quality of our manuscript. To address this point, we automatically quantified centrin and pericentrin/ γ -tubulin integrated densities as proxies for centriole length and PCM content, respectively, in two cell lines displaying centriole over-elongation. We then inferred the interdependency of PCM content and centriole length. As shown in Figures S10 B and C, both parameters are correlated in both cell lines therefore confirming that elongated centrioles recruit more pericentriolar material.

To further improve the analysis, we took advantage of the fact that mitoses were often visually displaying asymmetric centriole content in MDA-MB-435 cell line to directly compare, at the single cell level, the PCM recruitment ability of centrosomes containing overly-long centrioles versus centrosomes with normal-length centrioles. Please note that we used RPE-1 and MDA-MB-435 cells with only normal-length centrioles as controls. As shown in Figures 6A and S10A, mitoses containing overly long centrioles clearly display an asymmetry in the PCM content with poles with longer centriole(s) recruiting more than double the amount of PCM than the other poles. As a control, in RPE-1 and MDA-MB-435 mitotic cells displaying only normal-length centrioles, almost no difference in the total γ -tubulin and pericentrin intensities was observed between poles.

Altogether our results demonstrate that elongated centrioles recruit more pericentriolar material than normal length centrioles.

Minor points:

1). The enlarged photos in Fig. 1B should provide a scale bar.

We were grateful to reviewer for mentioning this missing scale bar. We modified the pictures accordingly by adding the aforementioned scale bars (see new Figure 1B).

2). Fig. 6C and Fig. S6C need to provide a scale bar.

We have repeated the experiments presented in Figure 6C following reviewer 2's recommendation. The new data are depicted in new Figure 6C and the images display scale bars.

We have also provided a scale bar for old Figure S6C which now became Figure S11A.

Reviewer #2

In this study Marteil and colleagues perform an extensive study of 60 cancer cell lines in terms of centrosome number and centriole size. This study is based on an automated image-analysis setup that allowed the authors to characterize centriole numbers and size in mitotic cells. The authors find that abnormal centrosome numbers and an elongated centriole size is a frequent characteristic of cancer cells. Based on their results they further propose that over-elongated centrioles can lead to centriole fragmentation, ectopic pro-centriole formation, and increase microtubule nucleation capacity, and that these defects lead to an elevated rate of chromosome segregation errors.

On the **plus side this study addresses an important biological question, which is the extent of centrosome abnormalities in cancer tissues**. While centrosomes are generally thought to be abnormal in cancer cells, **such a systematic study provides a much more definite answer**. Moreover the author describe **an novel type of centrosome defect, centriole over-elongation**,

and link it to the formation of multipolar spindles and chromosome segregation errors, which makes this study novel, original and of strong interest to the mitosis/centrosome field. On the negative side, this study suffers from a number of major weaknesses (see below), which partially undermine the major conclusions of this study. Therefore, even though this manuscript has a strong potential, it is too preliminary for publication, and it would benefit from a major set of revisions.

We were happy the Reviewer highlighted the importance of our systematic survey of centriole abnormalities in the NCI-60 panel, and the discovery of a novel type of defect, centriole over-elongation. We were also pleased the Reviewer considered our work novel and original with a strong potential of interest for a broad scientific community. However, Reviewer 2 considered our study too preliminary for publication, mostly due to the limited use of non-cancerous cell lines. We agreed with Reviewer 2 and we conducted several experiments to address these concerns.

Major points:

1.) Screening for centriole amplification: To characterize centriole amplification, the authors analyzed 50-60 cancer cells from breast, brain, colon, blood, lung, prostate, kidney, ovaries and skin and compared it to 1 control non-cancer cell line, the retina epithelial cells RPE-1. If at least 16% of the cells displayed more than 4 centrioles, cells were classified as having amplified centriole numbers. There are two fundamental problems with this screen. First, the sample size is very low. With a minimum of 50 cells and a cut-off probability of 16% one obtains a 95% confidence interval of +/- 10%, which means that the classification of many cell lines is uncertain. The sample size should be increased. Second, we do not know what is the variability of centriole numbers in non-cancer cells, since the authors only used 1 cell line (RPE-1) as a negative control. Moreover, this cell line originates from a tissue (retina), which is different from all the tested cancer cell line tissues. **Ideally, the authors should test one non-cancerous cell line from every tissue studied in this paper, as this would provide a comparison within the same tissue, and would define the variation in centriole numbers one should expect.** Improving these two points is essential, since the authors use this classification throughout the study.

We could obtain four non-cancerous cell lines from different tissues represented in the NCI-60 panel: HB2 (mammary luminal epithelial cells), HaCat (keratinocytes), LT97 (colon adenoma cells) and SAEC (small airway epithelial cells) and we added them to our study. We quantified centriole number and length in these non-cancerous cell lines using the same method as the one described for the secondary screening of the NCI-60 panel. The average percentage of cells with centriole amplification observed in the non-cancerous cell lines (RPE-1, HB2, HaCat, LT97 and SAEC) is 7% +/- 3, therefore confirming that non-cancerous cell lines display low level of centriole amplification with low variability among cell lines. We are now solely using these values to determine which cancer cell lines from the NCI-60 panel display significant increase in centriole number and, therefore, we have removed the previously used categories (low, moderate and high amplification). As shown in new Figure 2B, we set the cut-off for centriole amplification to 13% which corresponds to the average percentage of cells with centriole amplification in the non-cancerous-cell lines + 2 standard deviations.

In addition, we followed the same strategy to determine which NCI-60 cell lines exhibit significant centriole over-elongation. The average percentage of cells with centriole over-elongation observed in the non-cancerous cell lines (RPE-1, HB2, HaCat, LT97 and SAEC) is 1% +/- 2, therefore showing that the control of centriole length is very strong in non-cancerous cell lines. As shown in new Figure 3B, we set the cut-off for centriole over-elongation to 5% which corresponds to the average percentage of cells with centriole over-elongation in the non-cancerous-cell lines + 2 standard deviations.

We are grateful to reviewer 2 for this comment as it provides a more objective classification of the NCI-60 cell lines regarding centriole amplification and over-elongation. Furthermore, this new quantification provides important knowledge to the centrosome field as it provides the levels and variability of centriole amplification and over-elongation in a non-cancerous context.

2.) Link between centriole length and centriole amplification: the authors postulate the elongated centrioles might contribute to centriole amplification. **This is supported by EM-data, which show that elongated centrioles can fragment and/or lead to ectopic pro-centriole formation.** However, the crucial correlation between the two phenotypes is Figure 5A, which suggests that cells with elongated centrioles are more likely to also have an elevated number of centrioles. **However, if one takes the number given in Figure legend 5A for every category and performs a Chi-square test on a two-by-two table, one finds that correlation between elevated centriole length and centriole numbers is not statistically significant (p-value of 0.069 for a one-sided Chi-square test).** Although there is a tendency, the authors can on the basis of their numbers not conclude that an elongated centriole length leads to centriole amplification.

We appreciate the careful reading of our manuscript and thank the reviewer for bringing up this valid point. In the new version of our manuscript, we have removed the previously arbitrary established categories, therefore we have now investigated the interdependency of centriole amplification and over-elongation at the population level by calculating the Spearman correlation coefficient. As shown in new Figure S7, this coefficient is equal to 0.431 and $p < 0.01$, therefore suggesting that centriole amplification and elongation are correlated. To further strengthen this point, we tested if the proportion of long centrioles in cells with centriole amplification is statistically different from the expected proportion under the null hypothesis of independence (new Figure 5A). We found that it is indeed the case, therefore showing that centriole over-elongation and amplification are not independent.

To further test if centriole elongation directly induces centriole amplification, we over-expressed CPAP, a key promoter of centriole elongation, in two cancer cell lines from the NCI-60 panel with no significant increase in centriole number and length, T47D and SF268 (new Figure 5B). After 96 h, the percentage of mitotic cells displaying supernumerary centrioles is significantly increased upon CPAP overexpression therefore confirming that centriole elongation triggers amplification.

3.) Microtubule nucleation capacity: in Figure 6C and D the authors test whether the elongated centriole length leads to a higher microtubule nucleation capacity. Although the quantification in Figure 6D strongly supports this hypothesis, the picture shown in figure 6C is of poor quality, and one cannot recognize the typical microtubule aster that should emanate from the centrosomes in a microtubule nucleation assay. **The authors should therefore improve this assay to obtain much more convincing images reflecting microtubule nucleation.**

We repeated this assay using a different alpha-tubulin antibody that improved the MT staining (See new Figure 6C). We confirmed that overly long centrioles nucleate more MTs than normal-length centrioles. We hope that the quality of the new images will meet the quality criteria of *Nature Communications*.

4.) Centriole elongation and chromosome segregation: based on Figure 6E and F the authors propose that the elongated centrioles contribute to chromosome segregation errors. **The quantification in Figure 6F is convincing, but one worry is that the two phenotype are only linked by a correlation.** One could imagine that the MDA-MB-435 cell line is composed to two cell populations, one with elongated centrioles, the other without, and that the first population has a higher chromosome segregation error rate. The situation would be very different if the authors could suppress the segregation errors by reducing microtubule nucleation in these cells (gamma-tubulin or pericentrin depletion?).

We agreed with reviewer 2 for the need to substantiate the direct link between centriole over-elongation and chromosome segregation errors. Nevertheless, the experiment proposed by reviewer 2 is not feasible as we would have to selectively reduce the PCM level of overly long centrioles to the level of normal-length centrioles without reducing the PCM level of normal-length centrioles. Otherwise, we could be reducing the MT nucleation capacity of the normal-length centrioles rendering them less active, therefore mimicking centrosome loss, which induces chromosome segregation errors (Sir *et al*, 2013; Wong *et al*, 2015).

As an alternative and towards the same objective, we performed the gain of function experiment suggested by reviewer 1 (second part of major point 1). Briefly, we triggered centriole over-elongation in two cancer cell lines from the NCI-60 panel that exhibit low levels of centriole amplification and over-elongation, T47D and SF268, by transiently overexpressing CPAP, a major inducer of centriole elongation. We first observed that the proportion of prometaphases-metaphases displaying multipolar spindles, a known source of chromosome instability (Ganem *et al*, 2009; Silkworth *et al*, 2009), statistically increases upon CPAP overexpression in both cancer cell lines (Figure 6E). We then saw that the proportion of anaphase-telophase displaying chromosome segregation defects (polar asymmetry, DNA/chromosome bridge, lagging chromosomes, misaligned chromosomes and multipolar cells) is statistically enhanced upon CPAP over-expression in both cell lines (Figures 6F and S11C). Altogether, these results show that induced centriole elongation triggers chromosome segregation defects in cancer cell lines.

Moreover, it would be helpful to the reader **if the authors would quantify the type of errors seen in both cell populations**, since changes in microtubule dynamics are usually associated with lagging, merotelic chromosomes, and not so much with the chromosome bridges depicted in the immunofluorescence images of Figure 6E.

The different types of abnormalities are depicted in supplementary Figure S6 (now new Figure S11B).

Finally, it would be helpful if the authors **would speculate on possible mechanisms by which changes in microtubule nucleation might lead to chromosome segregation errors**.

We acknowledge that we did not sufficiently discuss this interesting point in the Discussion. We modified the text as follow (p18): "Finally, we showed that enlarged centrosomes enhance abnormal chromosome segregation, likely promoting aneuploidy. More precisely, we observed an increase in the occurrence of chromosome misalignments and polar asymmetry upon centriole over-elongation (both in MDA-MD-435 and upon CPAP over-expression in T47D and SF268 cells) and in the formation of lagging chromosomes in T47D expressing CPAP. These defects likely result from the formation of asymmetric mitoses with the pole containing longer centrioles, being over-active, capturing more chromosomes than the other pole. Consistent with this conclusion, a recent study demonstrated that mitoses with uneven numbers of daughter centrioles between the two poles, display asymmetric PCM content and, therefore MT nucleation capacities, phenotypes reminiscent of the ones we observed upon centriole over-elongation (Cosenza *et al*, 2017). This asymmetry in MT content leads to unequal kinetochore capture therefore increasing the rate of chromosome missegregation (Cosenza *et al*, 2017). Surprisingly, we also observed an increase in the occurrence of chromosome bridges upon centriole elongation. Chromosome bridges can arise from chromosome fusions, failed decatenation, or incomplete DNA replication. Overly long centrioles may induce this defect by triggering mitotic delays, therefore inducing Aurora B kinase-dependent telomere uncapping and chromosome fusion (Hayashi *et al*, 2012). Alternatively, chromosome bridges might be a consequence of the formation of lagging chromosomes. The latter can generate

micronuclei with fragmented chromosomes, which in the next cell cycle cause bridges (Zhang *et al*, 2015; Lindberg *et al*, 2008). Further investigations are now required to decipher how centriole elongation trigger chromosome segregation defects.

Reviewer #3 (Remarks to the Author):

The manuscript Deregulation of Centriole Length is Widespread in Cancer and Promotes Centriole Amplification and Chromosome Missegregation Abnormal numbers of centrioles and centrosomes and altered centrosome functions contribute to high rates of chromosome missegregation and are frequent in cancer. The authors of this manuscript decided to closely characterize this phenomenon by detailed analysis of centriole numbers and shape in the cancer cell line collection NCI-60. Their systematic survey of cancer cell lines revealed that many cells have high number of centrioles and many centrioles are abnormally elongated. These abnormalities correlated with the centrosome microtubule nucleation activity and with frequency of chromosome segregation error. **The work is carefully executed and the manuscript is well written, but lacks novelty and original insight. Most of the thoughts, ideas and findings from this manuscript have been previously shown, although not in 60 different cell lines. Therefore, this article seems appropriate for a more specialized journal, such as Cancer Research or Oncotarget.**

We appreciate that Reviewer 3 considers that our work has been carefully executed. However, we disagree that the study lacks sufficient novelty for *Nature Communications*, a point on which both reviewers 1 & 2 support our view. First, we emphasize the difference between conclusions based on the anecdotal analysis of a handful of cell lines, and an analysis based on a systematic and comprehensive study of a well- annotated collection such as the NCI-60, as we did here. The systematic analysis is more definitive and provides an important reference. Moreover, both reviewers 1 & 2 appreciated the novelty of our conclusions about centriole size and the fidelity of chromosome segregation in cancer cells. Finally, we feel that *Nature Communications* is very appropriate for this manuscript because it is not only relevant to cancer biologists, but also to cell biologists studying cell division. We have conducted several experiments/analyses to address Reviewer 3's remarks and we hope that Reviewer 3 will now consider our manuscript suitable for publication in *Nature Communications*.

Several remarks:

1. Although the authors look at multiple breast tumour samples, the main body of work has been done in culture cell lines. **The question to what degree is this informative about the real situation in cancer** is not even discussed.

Most of our work was performed in cancer cell lines, therefore we selected one tissue to validate our findings, and focused on breast carcinomas of the two different molecular sub-types represented in the panel (luminal and basal). As shown in Figure 4, we observed a strong correlation between the findings in cell lines and in patient samples. The types of tumours that show less amplification in the cell lines (Figure 2), are the same in the patient samples (Figure 4). These patient data supported the results of our systematic survey of centriole number deregulation in cancer cell lines, therefore supporting the use of the NCI-60 panel to study centriole abnormalities in cancer. We highlighted this in the results section (paragraph "Aggressive breast and colon cancer display high levels of centriole amplification") and in the discussion section (p15).

2. It is interesting that even the most affected cell lines show centriole amplification only in a minority of the cells. Why is that? How these cells differ? What causes the heterogeneity? This is not discussed at all either.

The most affected cell lines show more than 30% of their cells with centriole amplification (Figure 2). The wide heterogeneity is perhaps one of the most interesting points that rises from our work. We already had discussed this point in the previous version of our

manuscript, but we have now developed it further and with more emphasis. We now also discuss a recent study performed by Cosenza and collaborators: Discussion section, p16: "The large variability in the percentage of cells with supernumerary centrioles observed between cell lines, confirms the existence of an intrinsic centrosome number abnormalities "set point" for each cell line. This concept was initially proposed by Wong and collaborators. These authors generated cells without centrosomes in different cancer cell lines by treating them with a reversible inhibitor of centriole biogenesis. As cells keep on dividing without generating new centrosomes, the number of these structures is diluted within the population. After inhibitor washout, cells formed massive amounts of centrosomes *de novo*, but gradually recovered the initial level of centrosome amplification. This finding may reflect the existence of a tumour-specific dynamic equilibrium between stochastic emergence of cells with supernumerary centrioles and death of those cells. Fitting this model, a recent study followed cell fate after induced centriole over-duplication. During the first mitosis after centriole over-duplication, cells contained only two centrosomes, with at least one of them displaying multiple daughter centrioles. These cells formed normal mitotic bipolar spindles and subsequently divided into viable daughter cells. However, in the next cycle, centrioles disengaged, an event leading to the formation of supernumerary centrosomes. Here, cells either formed clustered or multipolar mitoses, both of which were mostly leading to unviable offspring therefore being negatively selected. In conclusion, while cells with multiple centrosomes may arise and be present in the population, part of their progeny may die. This population might survive, but only if having a certain degree of centriole amplification is beneficial to the overall population (e.g. by promoting increased ability to invade and/or yet unknown non-cell autonomous effects that promote survival, Ganier et al, BioRxiv). Further understanding of the "centrosome set point" is critical to successfully use centrosome amplification as a target for cancer therapy."

3. **Figures 6E – what is stained with the red signal on the right panel?** Is it CP110 as in the left panel? Why is it so different from the left panel?

The red signal on the right panel is indeed CP110 as seen on the insets. We acknowledge that sometimes the CP110 antibody also stains the midbody, as visible on some of the pictures depicted in new Figure 6D. We mentioned this in the figure legend.

4. The relationship between centriole changes and chromosome segregation errors is rather vague. It would be **more useful to look whether there is indeed a strong correlation of the centriole changes and CIN levels**. The number of chromosomes in karyotype and its range is known for the NCI60 cells. More careful analysis correlating the centriole numbers, chromosome numbers and the range and variance in the chromosome number might be more informative.

5. The same is true for "ploidy deregulation". This is a very vague term, as is the entire paragraph about ploidy and centriole numbers.

To address remarks 4 and 5 of reviewer 3, we have tested if centriole amplification correlates with the modal chromosome number, the number of numerical chromosome changes and the number of structural chromosomal rearrangement and do not observed any correlation with the aforementioned parameters, therefore confirming the absence of correlation between increase in ploidy, karyotypic complexity and centriole amplification. We have modified accordingly the paragraph about ploidy (p10).

6. How can the **altered centriole numbers cause anaphase bridges?**

This comment is not clear to us as we have only discussed the effect of long centrioles on chromosome segregation defects. However, we are now discussing possible mechanisms by which centriole over-elongation might lead to chromosome segregation errors (see discussion 18 and reply to major point 4 of reviewer 2).

Comment for all reviewers:

Please note that, using the classification based on the new threshold obtained from the quantifications in the non-cancerous cell lines and given the high proportion of cells with mutated p53 in the panel, the proportion of cell line displaying loss of p53 function in the group of cell lines with centriole amplification is not anymore statistically different from the proportion in the group of cell lines without significant centriole amplification. With our new classification, we are still finding that almost half of the p53 mutated cell lines do not show centriole amplification therefore reinforcing the idea that p53 loss is not sufficient to trigger centriole amplification.

We have modified our text accordingly and summarized the link between loss of p53 function and centriole amplification in the discussion section p17: "We showed that most cell lines with amplification lost p53 function; however, this event is not sufficient to trigger this abnormality, as almost half of the cell lines with deregulated p53 function do not show amplification. These results, together with the ones published by others, suggest that loss of p53 function is a prerequisite to sustain centrosome amplification across cancer, rather than a direct cause" (Coelho *et al*, 2015; Holland *et al*, 2012; Meraldi *et al*, 2002; Vitre *et al*, 2015; Marthiens *et al*, 2013; Sercin *et al*, 2016).

References:

Coelho, P.A. *et al*. Over-expression of Plk4 induces centrosome amplification, loss of primary cilia and associated tissue hyperplasia in the mouse. *Open biology* **5** (2015)

Cosenza MR, Cazzola A, Rossberg A, Schieber NL, Konotop G, Bausch E, Slynko A, Holland-Letz T, Raab MS, Dubash T, Glimm H, Poppelreuther S, Herold-Mende C, Schwab Y, Krämer A. Asymmetric Centriole Numbers at Spindle Poles Cause Chromosome Missegregation in Cancer. *Cell Rep*. 2017 Aug 22;20(8):1906-1920.

Ganem NJ, Godinho SA, Pellman D. A mechanism linking extra centrosomes to chromosomal instability. *Nature*. 2009 Jul 9;460(7252):278-82.

Ganier, O., Schnerch, D., Oertle, P., Lim, R., Plodinec, M., and Nigg, E.A., Structural centrosome aberrations promote non-cell-autonomous invasiveness. *BioRxiv*.

Hayashi, M.T., Cesare, A.J., Fitzpatrick, J.A., Lazzarini-Denchi, E. & Karlseder, J. A telomere-dependent DNA damage checkpoint induced by prolonged mitotic arrest. *Nat Struct Mol Biol* **19**, 387-394 (2012).

Holland, A.J. *et al*. The autoregulated instability of Polo-like kinase 4 limits centrosome duplication to once per cell cycle. *Genes & development* **26**, 2684-2689 (2012).

Lindberg, H.K., Falck, G.C., Jarventaus, H. & Norppa, H. Characterization of chromosomes and chromosomal fragments in human lymphocyte micronuclei by telomeric and centromeric FISH. *Mutagenesis* **23**, 371-376 (2008).

Marthiens, V. *et al*. Centrosome amplification causes microcephaly. *Nature cell biology* **15**, 731-740 (2013).

Meraldi, P., Honda, R. & Nigg, E.A. Aurora-A overexpression reveals tetraploidization as a major route to centrosome amplification in p53^{-/-} cells. *The EMBO journal* **21**, 483-492 (2002).

Sercin, O. *et al*. Transient PLK4 overexpression accelerates tumorigenesis in p53-deficient epidermis. *Nature cell biology* **18**, 100-110 (2016).

FUNDAÇÃO CALOUSTE GULBENKIAN
Instituto Gulbenkian de Ciência

Silkworth WT, Nardi IK, Scholl LM, Cimini D. Multipolar spindle pole coalescence is a major source of kinetochore mis-attachment and chromosome mis-segregation in cancer cells. *PLoS One*. 2009 Aug 10;4(8):e6564

Sir JH, Pütz M, Daly O, Morrison CG, Dunning M, Kilmartin JV, Gergely F. Loss of centrioles causes chromosomal instability in vertebrate somatic cells. *J Cell Biol*. 2013 Dec 9;203(5):747-56.
Vitre, B. *et al*. Chronic centrosome amplification without tumorigenesis. *Proceedings of the National Academy of Sciences of the United States of America* **112**, E6321-6330 (2015).

Wong YL, Anzola JV, Davis RL, Yoon M, Motamedi A, Kroll A, Seo CP, Hsia JE, Kim SK, Mitchell JW, Mitchell BJ, Desai A, Gahman TC, Shiau AK, Oegema K. Reversible centriole depletion with an inhibitor of Polo-like kinase 4. *Science*. 2015 Jun 5;348(6239):1155-60.

Zhang, C.Z. *et al*. Chromothripsis from DNA damage in micronuclei. *Nature* **522**, 179-184 (2015).

REVIEWERS' COMMENTS:

Reviewer #1 (Remarks to the Author):

Remarks to the author:

I have reread the current version and the rebuttal of the reviews. In their revised manuscript Marteil et al. have addressed many of my concerns, except the major one (see the major point 1). I found that the data they provided (e.g. recruit more PCM proteins, nucleate more microtubules, and enhance abnormal chromosome segregation) are mainly restricted to a single MDA-MB-435 cancer cell line. I requested the authors to perform similar experiments using two or more cancer cell lines with a feature of overly long centrioles. Unfortunately, in the rebuttal letter, the author stated that they were unable to obtain similar results in HOP-62 and H23 cell lines, due to an insufficient number of mitotic cells available for examination. Although the study of exogenous overexpression of CPAP could provide some useful information, it still represents an artificial condition. This is an important point. Since if only one cell line possesses these phenotypes, implying that the observed effects are not a common feature and restricted to only a minority of cancer cells. On the basis of this view, I am skeptical of the conclusion they have made in the revised manuscript. However, I will be happy to consider a future revision, if they could answer my major question listed above.

Reviewer #2 (Remarks to the Author):

The authors of the study "Over-elongation of Centrioles in Cancer promotes centriole amplification and chromosome missegregation" have performed an extensive round of revision to address the reviewers comments. Overall, the authors have done an excellent job and I fully support publication of the manuscript. In particular the additional analysis of several non-cancer cells and the CPAP overexpression experiments give much more weight to their conclusions. I have nevertheless a few minor comments, which are possible, but non-obligatory textual improvements.

1. When introducing the different classes of chromosome segregation errors, it might be helpful to explain some terms. In particular the term polar asymmetry is not well defined. I suspect the author mean a chromosome that was in the proximity of one of the two spindle poles at anaphase onset? Maybe defining this would help.
2. Along the same lines, the "misaligned DNA/chromosome" in telophase is not ideal, since unaligned chromosomes is rather a term for early mitosis. Maybe micronuclei or nuclear protrusion would be better suited?
3. Finally, in Figure S11B it is striking that overly long centrioles are mostly associated with an increase in chromosome bridges, a fact that is unexpected, which at the moment a bit hidden in the text. The author do a good job at speculating about the origin of these bridges, and I don't think that more mechanistic insight is necessary at this stage, but the study might benefit by highlighting this unexpected finding more.

Reviewer #4 (Remarks to the Author):

The manuscript by Marteil et al. describes a screen for centriole abnormalities in the NCI-60 panel of cell lines using elegant microscopy methods. In addition to the expected aberrations in centriole numbers, they also describe common deregulation in centriole length in a subset of cancer cell lines. Using a variety of cellular studies, they also report that overexpression of molecules that result in increased centriole length eventually triggers abnormal chromosome segregation suggesting a link with subsequent alterations in centriole numbers and chromosomal instability.

Overall, the technical quality of the paper is very high and I was positively surprised by the number of assays performed and controls used throughout the manuscript. After reading the answers to the former reviewers I realized that some of these assays were actually requested by the reviewers and I agree both in the convenience of including these assays and the way the authors have addressed these requests and previous criticisms.

Overall, I think the manuscript is of high technical quality and the information reported, basically a first comprehensive analysis of centriole abnormalities and their possible implications in genomic instability in a representative panel of cancer cell lines, is of interest for a general audience. The manuscript is quite complete in its current version and I recommend publication although there are a couple of points that I would like to discuss here in case the authors consider these comments useful.

I also agree with other reviewers that the assays performed to test the possible causal effect of centriole length in centriole number and chromosomal instability, while necessary and very well performed, are of limited reach as a complete demonstration of the relevance of centriole over-elongation in cancer. Most of the functional assays have been performed by overexpression of CPAP and we cannot formally rule out centriole elongation-independent functions. In addition, it is not clear to what extent CPAP overexpression is an oncogenic event in these cell lines (or primary tumors). Thus, while these assays indicate that CPAP overexpression can induce these effects, I would say that the relevance of centriole over-elongation inducing or modulating malignant transformation remains to be explored in detail in the future. Having said this, I would not consider this as a defect in the manuscript since a full demonstration will require further studies by different technologies including dedicated *in vivo* assays. The authors perhaps may want to revise the text to help readers to avoid possible over-conclusions.

The authors also indicate in the manuscript that "Our screen shows that centriole amplification is widespread and more prevalent in aggressive carcinomas" (abstract). I have selected this sentence as an example of the risk of using cell lines to generate conclusions applicable to primary tumors in patients. Most of the assays are performed in stable cell lines and it is well established that growing in culture alters many pathways in the cell being cell cycle regulation one of those. The number of tumor samples used in Figure 4 is low to generate strong conclusions (see also an additional comment on this figure below) and basal tumors are known to have specific deregulations (for instance pRb pathway) when compared to luminal tumors. Whereas the frequency of centriole aberrations may be higher in basal cancers (but please see comment below), it is still risky to generalize to "aggressive" tumors, whose definition (although difficult to establish in many papers) is based on proliferative and metastatic indexes not evaluated here. I understand that part of this discussion is purely semantic but the authors may want to revise the text to avoid conclusions that are beyond the data reported in the manuscript.

Nuclei shown in Fig. 4 suggest that centriole amplification correlates with polyploidy given the size of the nuclei shown in panel A. Whereas this does not modify the quantification shown in panel B and the conclusion that basal tumors may have increased number of centriole amplification, it would be interesting to test how strong is the correlation with polyploidy/aneuploidy (perhaps by quantifying nuclear volume). That may help in future studies in human samples. Do any of these tumors show obvious centriole over-elongation?

As the authors indicate it will be fantastic to find a signature of gene expression (from the published transcriptome studies of these cell lines) correlating with centriole over-elongation (and or amplification) but I understand this will require many validation studies beyond the scope of this manuscript.

Figure 7 is cited in pag. 12 before Fig. 6.

Page 28. Do the authors mean retractile index = 1.513 (instead of 1,513)?

We were very happy to receive your decision and we are pleased to send you a final revised version of our manuscript entitled "Over-elongation of Centrioles in Cancer Promotes Centriole Amplification and Chromosome Missegregation". We thank the reviewers for their helpful comments. We believe that we have now addressed the remaining concerns of the reviewers, and that our manuscript is ready for publication in *Nature Communications*.

Point by point answers to Reviewers

Reviewer #1 (Remarks to the Author):

I have reread the current version and the rebuttal of the reviews. In their revised manuscript Marteil et al. have addressed many of my concerns, except the major one (see the major point 1). I found that the data they provided (e.g. recruit more PCM proteins, nucleate more microtubules, and enhance abnormal chromosome segregation) are mainly restricted to a single MDA-MB-435 cancer cell line. I requested the authors to perform similar experiments using two or more cancer cell lines with a feature of overly long centrioles. Unfortunately, in the rebuttal letter, the author stated that they were unable to obtain similar results in HOP-62 and H23 cell lines, due to an insufficient number of mitotic cells available for examination. Although the study of exogenous overexpression of CPAP could provide some useful information, it still represents an artificial condition. This is an important point. Since if only one cell line possesses these phenotypes, implying that the observed effects are not a common feature and restricted to only a minority of cancer cells. On the basis of this view, I am skeptical of the conclusion they have made in the revised manuscript. However, I will be happy to consider a future revision, if they could answer my major question listed above.

We agree with Reviewer #1 that showing enhanced MTs nucleation and chromosome segregation defects in other cancer cell lines would have strengthen our conclusions. Nevertheless, we were able to show in an additional cell line displaying long centrioles, HOP-62, that centriole length and PCM content, using centrin and pericentrin/ γ -tubulin integrated densities as proxies, correlate. These results confirmed that elongated centrioles recruit more pericentriolar material. As previously mentioned, the number of mitotic cells with a clear and visible asymmetry in centriole content in this cell line was not sufficient to perform trustable quantifications of MT nucleation activity and chromosome missegregation. However, the results obtained by over-expressing CPAP in two cancer lines with no increase in centriole number and length, SF268 and T47D, greatly corroborate the results obtained in the cancer cell lines naturally displaying elongated centrioles, namely enhanced PCM recruitment and increased chromosomal instability. As a consequence, we are more confident now that centriole over-elongation leads to enhanced PCM recruitment and chromosomal instability

Reviewer #2 (Remarks to the Author):

The authors of the study "Over-elongation of Centrioles in Cancer promotes centriole amplification and chromosome missegregation" have performed an extensive round of revision to address the reviewers comments. Overall, the authors have done an excellent job and I fully support publication of the manuscript. In particular the additional analysis of several non-cancer cells and the CPAP overexpression experiments give much more weight to their conclusions. I have nevertheless a few minor comments, which are possible, but non-obligatory textual improvements.

We were pleased that Reviewer #2 considered that our manuscript is now suitable for publication. We are thankful to Reviewer #2 for his/her useful remarks that we have now addressed.

1. When introducing the different classes of chromosome segregation errors, it might be helpful to explain some terms. In particular the term polar asymmetry is not well defined. I suspect the author mean a chromosome that was in the proximity of one of the two spindle poles at anaphase onset? Maybe defining this would help.

We agree that we did not sufficiently define the category "polar asymmetry". Following Reviewer 2's suggestion, we have now defined it as "chromosome(s) in the vicinity of one of the two spindle poles at anaphase onset" and have modified the text accordingly (see page 14 of supplementary material and page 24 of the manuscript).

2. Along the same lines, the "misaligned DNA/chromosome" in telophase is not ideal, since unaligned chromosomes is rather a term for early mitosis. Maybe micronuclei or nuclear protrusion would be better suited?

We agree that "misaligned DNA/chromosome" is more suitable for metaphase and we have replaced this term by "DNA/chromosome protrusion" (see page 14 of supplementary material, Supplementary Figure 11 and page 24 of the manuscript).

3. Finally, in Figure S11B it is striking that overly long centrioles are mostly associated with an increase in chromosome bridges, a fact that is unexpected, which at the moment a bit hidden in the text. The author do a good job at speculating about the origin of these bridges, and I don't think that more mechanistic insight is necessary at this stage, but the study might benefit by highlighting this unexpected finding more.

We have now clearly mentioned in the text that the increased incidence of chromosome segregation defects was especially due to an increase in DNA/chromosome bridge in MDA-MB-435 cell line (page 11).

Reviewer #4 (Remarks to the Author):

The manuscript by Marteil et al. describes a screen for centriole abnormalities in the NCI-60 panel of cell lines using elegant microscopy methods. In addition to the expected aberrations in centriole numbers, they also describe common deregulation in centriole length in a subset of cancer cell lines. Using a variety of cellular studies, they also report that overexpression of molecules that result in increased centriole length eventually triggers abnormal chromosome segregation suggesting a link with subsequent alterations in centriole numbers and chromosomal instability. Overall, the technical quality of the paper is very high and I was positively surprised by the number of assays performed and controls used throughout the manuscript. After reading the answers to the former reviewers I realized that some of these assays were actually requested by the reviewers and I agree both in the convenience of including these assays and the way the authors have addressed these requests and previous criticisms.

Overall, I think the manuscript is of high technical quality and the information reported, basically a first comprehensive analysis of centriole abnormalities and their possible implications in genomic instability in a representative panel of cancer cell lines, is of interest for a general audience. The manuscript is quite complete in its current version and I recommends publication although there are a couple of points that I would like to discuss here in case the authors consider these comments useful.

We were very pleased that Reviewer #4 liked our manuscript and considered it suitable for publication. We are thankful to Reviewer #4 for his/her useful comments that we have now addressed.

I also agree with other reviewers that the assays performed to test the possible causal effect of centriole length in centriole number and chromosomal instability, while necessary and very well performed, are of limited reach as a complete demonstration of the relevance of centriole over-elongation in cancer. Most of the functional assays have been performed by overexpression of CPAP and we cannot formally rule out centriole elongation-independent functions. In addition, it is not clear to what extent CPAP overexpression is an oncogenic event in these cells lines (or primary tumors). Thus, while these assays indicate that CPAP overexpression can induce these effects, I would say that the relevance of centriole over-elongation inducing or modulating malignant transformation remains to be explored in detail in the future. Having said this, I would

not consider this as a defect in the manuscript since a full demonstration will require further studies by different technologies including dedicated in vivo assays. The authors perhaps may want to revise the text to help readers to avoid possible over-conclusions.

We have indeed solely used CPAP overexpression to strengthen the direct link between centriole elongation and chromosomal instability and therefore we cannot exclude centriole elongation-independent functions of CPAP overexpression. However, the results obtained with this strategy corroborate well the results obtained in the cancer cell lines naturally displaying elongated centrioles (MDA-MB-435 and HOP-62), namely enhanced PCM recruitment and increased chromosomal instability. As a consequence, we are more confident now that centriole over-elongation leads to enhanced PCM recruitment and chromosomal instability. That said and in line with Reviewer 4, we agree that our manuscript does not address the role of centriole over-elongation in malignant transformation. Following Reviewer 4 remarks, we have modified the text to avoid over-interpreting the data (changes are highlighted in yellow):

- Page 11-12: "In conclusion, our results suggest that overly long centrioles induce the formation of over-active centrosomes which enhance chromosome missegregation, both directly and indirectly via centrosome amplification. This likely leads to aneuploidy and therefore **may** participate in tumourigenesis.
- Page 15: **Altogether, our work demonstrates that centriole over-elongation enhances microtubule nucleation and chromosomal instability, two known tumourigenic features. Further studies are now required to determine if centriole over-elongation is sufficient to trigger tumourigenesis in vivo.**

The authors also indicate in the manuscript that "Our screen shows that centriole amplification is widespread and more prevalent in aggressive carcinomas" (abstract). I have selected this sentence as an example of the risk of using cell lines to generate conclusions applicable to primary tumors in patients. Most of the assays are performed in stable cell lines and it is well established that growing in culture alters many pathways in the cell being cell cycle regulation one of those. The number of tumor samples used in Figure 4 is low to generate strong conclusions (see also an additional comment on this figure below) and basal tumors are known to have specific deregulations (for instance pRb pathway) when compared to luminal tumors. Whereas the frequency of centriole aberrations may be higher in basal cancers (but please see comment below), it is still risky to generalize to "aggressive" tumors, whose definition (although difficult to establish in many papers) is based on proliferative and metastatic indexes not evaluated here. I understand that part of this discussion is purely semantic but the authors may want to revise the text to avoid conclusions that are beyond the data reported in the manuscript.

We agree with Reviewer 4 and have now modified the text accordingly (changes are highlighted in yellow):

Page 2: "Our screen shows that centriole amplification is widespread in **cancer cell lines** and highly prevalent in aggressive **breast** carcinomas".

Page 8: "Our results **suggest** that centriole amplification is more prevalent in specific subtypes of breast and colon cancer, which are both associated with chromosome instability and worse prognosis".

"These results suggest that centriole amplification specifically occurs in more aggressive molecular **breast** tumour subtypes. (we removed "explaining the diversity of data observed within each tissue").

Page 13: Our survey established that centriole over-elongation and amplification are widespread in cancer, **the latter correlating with aggressiveness in breast and colon cancer cell lines.**

Page 13: "We showed that centriole amplification is more prevalent in specific molecular subtypes of breast (basal-like) and colon (CIN molecular subtype) **cancer cell lines** which represent particularly aggressive carcinomas, associated with poor prognosis. Specific molecular features preferentially present in these carcinomas **may** underlie the presence of high levels of centrosome amplification".

Nuclei shown in Fig. 4 suggest that centriole amplification correlates with polyploidy given the size of the nuclei shown in panel A. Whereas this does not modify the quantification shown in panel B and the conclusion that basal tumors may have increased number of centriole amplification, it would be interesting to test how strong is the correlation with polyploidy/aneuploidy (perhaps by quantifying nuclear volume). That may help in future studies in human samples. Do any of these tumors show obvious centriole over-elongation?

The reviewer makes an important point, as ploidy may be an important prognostic factor for breast cancer and basal-like tumours that are considered more 'aggressive'. However, the evaluation of ploidy by volume quantification in these breast tumour samples is not completely accurate since some of the nuclei are only partially observed due to the tissue section technique, therefore introducing a bias in the quantification.

However, we agree that this point may be interesting for future studies in human samples, which we would preferably do with fresh tissue specimens.

Regarding centriole over-elongation in tumour samples, we are currently unable to address this point. We do not have yet a good experimental readout for centriole length in tumour samples, despite several trials with different antibodies. Similar to ploidy, we hope to address this point in a near future using fresh tissue specimens.

As the authors indicate it will be fantastic to find a signature of gene expression (from the published transcriptome studies of these cell lines) correlating with centriole over-elongation (and or amplification) but I understand this will require many validation studies beyond the scope of this manuscript.

In accordance with reviewer 4, we believe that this suggestion is beyond the scope of this manuscript and will be the focus of future studies.

Figure 7 is cited in pag. 12 before Fig. 6.

We have removed the citation of Figure 7 in page 12 (current page 10).

Page 28. Do the authors mean retractile index = 1.513 (instead of 1,513)?

We have now replaced 1,513 by 1.513.